# A novel probabilistic source apportionment approach: Bayesian Auto-correlated Matrix Factorization

Anton Rusanen[1,2], Anton Björklund[3], Manousos Manousakas[4], Jianhui Jiang[4,5], Markku T. Kulmala[1,6,7], Kai Puolamäki[1,3], and Kaspar R. Daellenbach[4]

[1]Institute for Atmospheric and Earth System Research (INAR) / Physics, Faculty of Science, University of Helsinki, Finland
[2]Atmospheric Composition Research, Finnish Meteorological Institute, Helsinki, Finland
[3]Department of Computer Science, Faculty of Science, University of Helsinki, Finland
[4]Laboratory of Atmospheric Chemistry, Paul Scherrer Institute (PSI), 5232 Villigen-PSI, Switzerland
[5]Shanghai Key Lab for Urban Ecological Processes and Eco-Restoration, School of Ecological and Environmental Sciences, East China Normal University, 200241, Shanghai, China
[6]Aerosol and Haze Laboratory, Beijing Advanced Innovation Center for Soft Matter Sciences and Engineering, Beijing University of Chemical Technology (BUCT), Beijing, China
[7]Joint International Research Laboratory of Atmospheric and Earth System Sciences, School of Atmospheric Sciences, Nanjing University, Nanjing, China

**Correspondence:** Anton Rusanen (anton.rusanen@helsinki.fi), Kaspar R. Daellenbach (kaspar.daellenbach@psi.ch)

**Abstract.**

The concentrations of atmospheric particulate matter and many of its constituents are temporally auto-correlated. However, this information has not been utilised in source apportionment methods. Here, we present a Bayesian matrix factorization model (BAMF) that considers the temporal auto-correlation of the components (sources) and provides a direct error estimation. The performance of BAMF is compared to positive matrix factorization (PMF) using synthetic Time-of-Flight Aerosol Chemical Speciation Monitor data, representing different urban environments from typical European towns to megacities. We find that BAMF resolves sources with overall higher factorization performance (temporal behaviour and bias) than PMF on all datasets with temporally auto-correlated components. Highly correlated components continue to be challenging and ancillary information is still required to reach good factorizations. However, we demonstrate that adding even partial prior information about the chemical composition of the components to BAMF improves the factorization. Overall, BAMF-type models are promising tools for source apportionment and merit further research.

## 1 Introduction

Air pollution in the form of particulate matter (PM) has a substantial impact on earth's climate (IPCC, 2021) and severe adverse effects on human health (Lelieveld et al., 2015; Daellenbach et al., 2020). PM's noxiousness could strongly depend on the particles' chemical composition, which is governed by their origin (Bates et al., 2019; Daellenbach et al., 2020). PM is affected by many emission sources and dynamic atmospheric processes, making PM a poorly understood complex mixture, especially the organic aerosol (OA) fraction of PM. Typically directly emitted OA (primary OA - POA) is distinguished from OA formed in the atmosphere from emitted vapours by nucleation or condensation (secondary OA - SOA). Identifying and

quantifying the sources of PM is, therefore, essential for designing effective and efficient air pollution reduction strategies.
Such analyses (called source apportionment) combine chemical characterization data with non-negative matrix factorization methods. The idea is to use the variation in the chemical composition of a set of measurements, such as outputs from mass spectrometers, to decompose the measurements into "source terms" using non-negative matrix factorization. The underlying assumption is that the measurement is a linear combination of strictly non-negative source terms.

Multiple methods for weighted non-negative matrix factorization exist (Wang and Zhang, 2012). A widely used method in atmospheric sciences is positive matrix factorization (PMF) (Paatero and Tapper, 1994), which has been used in over a thousand papers (Hopke, 2016). In earlier studies, chemical mass balance, CMB, was a popular method, but it has the drawback that factor profiles must be defined beforehand, see e.g. Watson et al. (2001). This introduced significant uncertainty since these factor profiles are usually not known beforehand or only with considerable uncertainty. PMF improved on this by optimizing the source profiles (Canonaco et al., 2013).

Previous studies have revealed that chemical data from the Aerosol Mass Spectrometer family (Aerodyne Aerosol Mass Spectrometer (Canagaratna et al., 2007), Aerosol Chemical Speciation Monitor (Ng et al., 2011; Fröhlich et al., 2013)) retains sufficient information for the resolution of some sources (Zhang et al., 2007, 2011; Daellenbach et al., 2017). However, distinguishing factors with chemical or temporal similarities or accurately resolving low-concentration factors is often challenging (Ulbrich et al., 2009; Canonaco et al., 2013; Zhang et al., 2011; Heikkinen et al., 2021). Several studies have shown that utilizing a priori information to constrain POA sources' chemical composition or time series is usually required to accurately estimate their contribution to OA (Canonaco et al., 2013; Crippa et al., 2014; Reyes-Villegas et al., 2016; Schlag et al., 2017; Zhang et al., 2018; Huang et al., 2019; Zhu et al., 2018; Chazeau et al., 2022). In addition, different statistical data reduction methods applied to mass spectrometry data extract different components (Isokääntä et al., 2020). This demonstrates that the problem does not have one unique solution, and the choice of method can emphasize different features of the resolved components.

While developments related to source apportionment, in atmospheric science, focused on different ways to pre- and post-process data (Zhang et al., 2019), the underlying solver algorithm mainly remained the same: PMF. Rolling PMF (Canonaco et al., 2021) refers to a pre-processing strategy feeding only subsets of data (e.g., 7 or 14 days) to the PMF solver. This allows for a temporal variation of the chemical composition of sources (particularly relevant for SOA), even if their profiles remain static within each PMF run (Canonaco et al., 2021).

The commonly used optimization goal Q in PMF only accounts for reconstruction of the data (Wang and Zhang, 2012; Paatero and Tapper, 1994). It lacks time information, which is a drawback considering some atmospheric measurements exhibit strong temporal auto-correlation (see, e.g., Figure 1). Earlier studies have also found sources with longer cycles due to emissions, such as traffic, and meteorological conditions (Daellenbach et al., 2020; Chen et al., 2022). Here, we present a probabilistic matrix factorization method that accounts for auto-correlation. We evaluate the model's performance in resolving air pollution sources based on realistic synthetic chemical data.

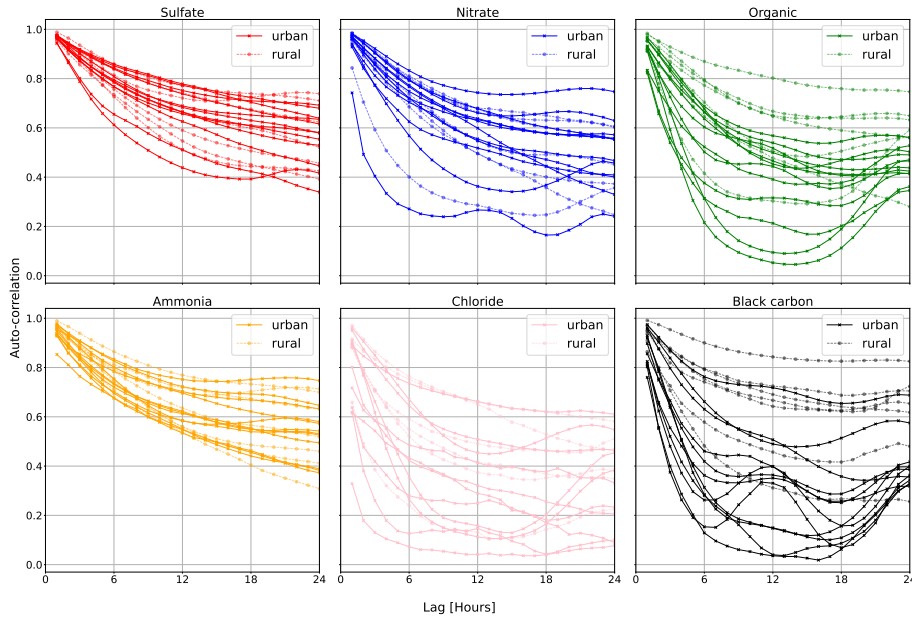

**Figure 1.** The auto-correlations of the hourly means of several measured aerosol constituents at 19 different sites in Europe. The auto-correlation is the Pearson correlation coefficient between the original and the delayed time series. Auto-correlations at lag of 1 and 2 hours are very high in most cases. This data shows that particulate matter constituents exhibit strong lag-1 auto-correlation, consistent with earlier such statements in literature (e.g., Hirtzel et al. (1982)). Auto-correlations calculated on data from Chen (2022).

## 2 Methods

In this section we will discuss the methods used in the paper, starting with notation in Section 2.1. We define the BAMF model in Section 2.2 and how we find solutions in Section 2.3 using the pre- and postprocessing steps in Section 2.4. In Section 2.5 we describe how we compare BAMF to PMF, which is defined in Section 2.6.

### 2.1 Notation

In this paper, we describe the data by $\mathbf{X} \in \mathbb{R}^{n \times m}$, where the rows $i \in [n] = \{1, \ldots, n\}$ correspond to measurements taken at consecutive times $t_i$. The columns $j \in [m] = \{1, \ldots, m\}$ correspond to the different dimensions of the measurement. Our objective is to find a lower dimensional non-negative decomposition $\mathbf{X} \approx \mathbf{GF}$ with $p$ factors such that $\mathbf{G} \in \mathbb{R}_{\geq 0}^{n \times p}$ and $\mathbf{F} \in \mathbb{R}_{\geq 0}^{p \times m}$, where $p \ll \min(n, m)$. In other words, the objective is to present the data as a multiplication of two much smaller matrices. The rows of $\mathbf{F}_{i \cdot}$ contain the time-independent components of the decomposition, which we call *factor profiles*. Factor profiles are defined to sum to unity to facilitate comparisons between different datasets and models. The columns of $\mathbf{G}_{\cdot i}$ contain the time dependency of each of the rows of $\mathbf{F}$; we will call these the *factor time series*. Simply put, the factor

profiles represent the sources' concentration and time independent chemical composition, and the factor time series describes the sources' time-dependent concentration. Note that the ordering of these profiles is arbitrary for the overall solution.

## 2.2 Bayesian Auto-correlated Matrix Factorization, BAMF

We define a Bayesian probabilistic model that captures our prior assumptions of the process that generated the measurements. The only observed variables in our model are the data matrix $\mathbf{X} \in \mathbb{R}^{n \times m}$ and the uncertainty estimate $\boldsymbol{\sigma} \in \mathbb{R}_{\geq 0}^{n \times m}$. Together they define the probability distribution of the observed concentration of each m/z at any point in time with the data matrix being the average and the uncertainty matrix the standard deviation of the distribution, here the Gaussian distribution. In addition to these observed variables, there are several latent variables. These include the matrices $\mathbf{G}$ and $\mathbf{F}$ mentioned above, vectors $\boldsymbol{\alpha} \in \mathbb{R}_{\geq 0}^{p}$ and $\boldsymbol{\beta} \in \mathbb{R}_{\geq 0}^{p}$ which determine the auto-correlation behaviour of the model, as well the "noise-free data matrix" $\mathbf{Z} \in \mathbb{R}_{\geq 0}^{n \times m}$. We define the probabilistic model as

$$\boldsymbol{Z} \quad = \quad \boldsymbol{GF} \tag{1}$$

$$\boldsymbol{X}_{ij} \quad \sim \quad \text{Normal}(location = \boldsymbol{Z}_{ij}, \; scale = \boldsymbol{\sigma}_{ij}) \text{ for all } i \in [n] \text{ and } j \in [m] \tag{2}$$

$$\boldsymbol{F}_{i\cdot} \quad \sim \quad \text{Dirichlet}(\mathbf{1}_m) \text{ for all } i \in [p] \tag{3}$$

$$\boldsymbol{G}_{i+1,k} \quad \sim \quad \text{Cauchy}(location = \boldsymbol{G}_{ik}, \; scale = \boldsymbol{\alpha}_k \Delta t_i + \boldsymbol{\beta}_k) \text{ for all } k \in [p] \text{ and } i \in [n-1] \tag{4}$$

where Normal corresponds to the normal probability distribution with a given mean and standard deviation, and Dirichlet to the Dirichlet distribution parameterized by a unit vector $\mathbf{1}_m$. The model specification implies that the components of $\mathbf{F}_{i\cdot}$ can have values from $[0,1]$ with equal likelihood, but all rows must sum to unity. Cauchy is the Cauchy probability distribution, with the width depending on the time difference between the $i$th and $(i+1)$th observation ($\Delta t_i = t_{i+1} - t_i$). Essentially our model describes the data as a non-negative matrix decomposition (NMF) with a lag-1 auto-correlation term and a Gaussian error term for the reconstruction of $\mathbf{X}$.

We chose the Cauchy distribution for the auto-correlation term because the long tails make large jumps between the $i$th and $(i+1)$th observation more probable than, for example, a Gaussian distribution. Other choices are possible, but the experiments in this paper suggest the Cauchy distribution works as an approximation for real data. Choosing the distribution shape also implicitly influences the weight of $\mathbf{G}$'s auto-correlation. The $\boldsymbol{\alpha}_k \Delta t_i + \boldsymbol{\beta}_k$ determines the scale of the Cauchy distribution. The $\alpha$-terms allow the model to deal with time steps of different lengths and missing data, since it forms a simple linear model for the scale. It has a minimum width $\boldsymbol{\beta}_k$ and increases linearly as $\Delta t$ increases (with $\boldsymbol{\alpha}_k$). Thus, for arbitrarily large time steps the Cauchy approaches a uniform distribution. In physical terms this means that at short timesteps we expect the values of $\mathbf{G}$ to stay close to the previous value and at large timesteps larger deviations have higher probability. It is possible to use other formulations for the width, which would be appropriate if one wishes to include a more complex and computationally intensive description of auto-correlation. It is also possible to consider more than lag-1 auto-correlation.

### 2.2.1 Uncorrelated Bayesian matrix factorization, BAMF-0

For comparison, we created a version of the BAMF model without the lag-1 auto-correlation terms of Equation (4). In other words, the model consist entirely of Equations 1-3. The model is otherwise identical to the BAMF model. This variation is essentially a probabilistic weighted NMF model, making it possible to assess the impact of the auto-correlation terms on the solution.

### 2.2.2 Bayesian matrix factorization with additional constraints, BAMF-C

In source apportionment analyses, it is common to utilize reference spectra as boundary conditions for the factor analysis – to find components with, e.g., previously observed chemical compositions. We include this scenario in another model, called BAMF-C, by adding peak intensity ratios to the BAMF model,

$$F[ii, jj]/F[kk, ll] \sim \text{Normal}(\text{ratio}, \text{width}) \tag{5}$$

where $ii, jj$ and $kk, ll$ are the indices of matrix $\mathbf{F}$ indicating the peak pair to constrain. The ratio is the desired intensity ratio, and width is a free parameter describing the width of the distribution, i.e. the uncertainty of the intensity ratio. This approach allows constraining the range of $\mathbf{F}$ for arbitrarily many $m/z$ pairs, which is currently not done in the other models. A similar concept could constrain $\mathbf{G}$ to have a similar time behaviour as an external ancillary measurement, e.g., of a source tracer. The constraint is similar to the widely used *a-value* (anchor value) approach in Source Finder coupled to PMF (Canonaco et al., 2013, SoFi/PMF) with two notable differences. Firstly, the intensity ratio of the peaks is constrained in BAMF-C, while the a-value constraint approach in SoFi/PMF uses the peak intensity. Secondly, BAMF-C has a soft-boundary, with increasing penalty term as distance from the anchor grows. SoFi/PMF employs a hard boundary (defined as a relative deviation from the a-value), without additional penalty on the object function Q for deviation from the anchor. In the present study, we evaluate the performance of the PMF and BAMF algorithm itself without discarding sub-optimal solutions (PMF) or samples (BAMF) during post-processing. Appendix F lists the profiles used for constraints in this work.

## 2.3 Solver

We use Stan (Carpenter et al., 2017) to compile and run the probabilistic models. Stan solves the probabilistic inference problem with a Markov Chain Monte Carlo (MCMC) method. Instead of obtaining a single solution for the latent variables, for example, by finding the model with the highest likelihood, we get an empirical distribution of possible solutions from which we can infer, e.g., confidence intervals. Stan takes our model and observed variables as input and outputs samples from the posterior distribution of the latent variables. We run multiple MCMC chains, starting from different initial conditions and usually extract a few thousand posterior samples per chain.

The standard way to initialize the model in Stan is by randomly sampling from the prior distributions. However, our model has many parameters with fairly strict distributions. Consequently, we found this starting point to be poor, sometimes causing Stan to markedly slow down, or even fail. Hence, we initialize the model with a point solution. We utilise Stan's capability to

find a single maximum-a-posteriori (MAP) point solution for the parameters, which we use as the initialization. Note, however, that the solutions typically have several local optima, in which case the point solution is only one such local optimum.

### 2.3.1 Hamiltonian Markov Chain Monte Carlo

Stan (Carpenter et al., 2017) uses a Hamiltonian Markov Chain Monte Carlo method to draw samples from the posterior distribution of our model (Equations 1-4) given the data. We go through the basic idea here, but direct readers to (Carpenter et al., 2017; Gelman et al., 2014) and references therein for a more detailed explanation.

The samples are drawn in proportion to the posterior probability of each sample. Obtaining samples from a multidimensional posterior distribution is a non-trivial task. For effective sampling, we use Hamiltonian Monte Carlo (HMC), a method where the gradient of the distribution and an ancillary variable called momentum are used to direct the chain to explore the typical set (Gelman et al., 2014). Specifically, we use the HMC based No-U-Turn Sampler (NUTS) sampler (Hoffman and Gelman, 2014) from Stan. Stan uses warmup iterations to estimate the parameters the NUTS sampler needs before drawing the posterior samples used in the computations (Carpenter et al., 2017).

The maximum-a-posteriori (MAP) estimate is used as a starting point for the sampling. It is found with an optimization method on the same probability distribution used by the sampling. We use the LBFGS (Liu and Nocedal, 1989) gradient-based optimization algorithm included in Stan for MAP estimates (Carpenter et al., 2017).

### 2.4 Pre- & post-processing

Before running our model, we normalize the data such that the mean of the data ($\mathbf{X}$) is 1. The error estimate is scaled with the same scaling factor. The equations to do this are

$$f_{norm} = \sum_{i,j} \boldsymbol{X}_{ij}/(n \times m)$$

$$\boldsymbol{X}_{ij}^* = \boldsymbol{X}_{ij}/f_{norm}$$

$$\boldsymbol{\sigma}_{ij}^* = \boldsymbol{\sigma}_{ij}/f_{norm}$$

where $f_{norm}$ is a scalar normalization factor, $\boldsymbol{X}_{ij}^*$ and $\boldsymbol{\sigma}_{ij}^*$ are the scaled data and error, respectively, which we use as model inputs. This normalization is done so that we can use a consistent scale for priors and posteriors, making the modelling easier, and the denormalization is performed to return the results to familiar units. The normalization is optional, a user can also choose to use non-scaled values.

Stan outputs posterior samples from the two matrices $\mathbf{F}$ and $\mathbf{G}$, representing the factors' time-independent chemical composition and their time-dependent concentration. Since all the magnitude information is in $\mathbf{G}$, $\mathbf{G}$ needs to be renormalized by simply multiplying with the normalization factor. The rows of $\mathbf{F}$ are constrained to sum to unity, and are, thus, directly comparable to mass spectrometric references, which are normalized similarly (Crippa et al., 2013; Ulbrich et al., 2022).

### 2.4.1 Sorting the components

The order of the components in **F** and **G** is arbitrary in our samples. The problem is not unique to BAMF but inherent to all such matrix decompositions. To be able to compare solutions, we need to be able to sort the components. The contribution of the same component to **Z** should be similar between two samples. To calculate the contribution for each component, we multiply the row of **F** with the corresponding column of **G** and use this to sort the components.

To select the ordering of the components, we take a small number of representative samples, usually the last five, and compute the optimal permutation using the Hungarian algorithm (Kuhn, 1955), which is a cost minimization algorithm which minimizes the cost of assigning values. In this case we are minimizing the Manhattan distances, which is the sum of absolute differences between the **Z** contributions in the samples. We then select the most common permutation as the ordering of the factors for all samples.

We use this approach for sorting the outputs of all models (BAMF-0/C, BAMF, PMF) to ensure the most direct comparability of the results. Finally, median, 25% and 75% percentiles are computed using the sorted samples. In the comparisons, we use medians for all models, but in some figures, we also show 25% and 75% percentiles. The median, or any central estimate, is not guaranteed to be the "best" optimized solution in any metric (probability or root mean squared sum of residuals). Still, we use it to represent a reasonable solution inferred from the samples.

### 2.5 Evaluating model performance

The first metric to check is if the model explains the data well (*reconstruction performance*). If **X** is not reconstructed appropriately, the solution is not acceptable. This can either mean that the data cannot be factorized this way (the model assumptions are wrong) or that the solver failed to find a solution. In such cases, the number of iterations should be increased, or solver parameters must be adjusted, such as the number of warm-up samples and parameters influencing the step size.

Even at moderate data sizes, assessing if the original data falls within the model's confidence bounds for every variable individually is not practical. Therefore, we summarised this information by computing the model residual (difference between the model input and output, mean of all samples) normalized with the uncertainty of the model output (standard deviation of all samples); essentially observing if the original data is inside the sample standard deviation.

$$\boldsymbol{S}_{ij} = (\boldsymbol{X}_{ij} - E[\boldsymbol{X}_{samples}])/\sigma(\boldsymbol{X}_{samples}) \tag{6}$$

Data in $\boldsymbol{S}$ should be centred at zero and have a standard deviation below 1, which means the data is often less than 1 model standard deviation away from the model mean. In addition, we also use a common evaluation metric in PMF analyses ($Q_m/Q_{exp}$), where $Q_m$ is defined as (Canonaco et al., 2013, notation adapted):

$$Q_m = \sum_{i,j} \left( \frac{(\boldsymbol{X} - \boldsymbol{Z})_{i,j}}{\boldsymbol{\sigma}_{i,j}} \right)^2 \tag{7}$$

Essentially $Q_m$ describes the sum of squared model residuals normalized to the input error. We use the same reduction as Zhang et al. (2011) where $Q_{exp}$ is approximated as data size and denote $Q_m/Q_{exp}$ as $Q_m*$.

For synthetic data—with a known ground truth—it is possible to assess how well the methods resolve the actual components in addition to the reconstruction performance. We call this evaluation *factorization performance*. We compare the median solutions with the corresponding actual components by calculating the average distance, Pearson and Spearman (nonlinear) correlations. Optimal matching between median solutions and true components is obtained using the Hungarian algorithm (Kuhn, 1955). The approach is similar to the sorting above but with the true components defining the order. Direct comparison to true components is only possible in cases where the number of true components matches the number of modelled components. Otherwise, the model must combine multiple components or create additional ones.

## 2.6 PMF

We use PMF, specifically the multilinear engine 2 (ME-2) controlled by the user interface SoFi (Canonaco et al., 2013; Paatero and Tapper, 1994), as a baseline comparison. PMF solves the decomposition in Equation 1 by minimizing the sum of the squared residuals normalized with the input error see Equation 7 (object function), given the boundary condition that all values must be positive (Canonaco et al., 2013). Since SoFi finds local optima, we ran it with different random seeds to get multiple solutions for all comparisons. As the runs have varying starting points, they often lead to different local optima, especially in cases with high rotational ambiguity. Thus, PMF provides a collection of local minima, while BAMF tries to sample the model's posterior distribution, including plausible answers that are not minima. For simplicity, we will refer to the group of PMF solution sets as samples, even though PMF is not a sampler.

A priori information in the form of known rows of factor profiles or of known columns of factor time series can be added to the model to reduce the rotational ambiguity. By adding this external information, the user can reduce the space PMF searches for the optimized solution, reducing the rotational ambiguity of the solution. Using external data to run PMF is usually referred to as constraining the solution, and external information is used as constraints. Here we used two approaches, a) entirely unconstrained PMF runs and b) constrained runs using external source profiles. We rely on the commonly used a-value approach to constrain the PMF runs. In the a-value approach, the user inputs one or more factor profiles or factor time series and defines a relative tolerated deviation from the anchor (termed a-value) (Canonaco et al., 2013). Constraint strengths are not directly comparable between the hard cut-off approach used in PMF and the softer Gaussian error term approach used in BAMF-C. For the best possible comparability, we first ran BAMF-C (constraint strength 0.001) and determined the equivalent a-value by taking the maximum deviation from the anchor value on a constrained component in $\mathbf{F}$ (a-value of $18\%$). See Appendix F for the profiles used to make this comparison.

## 3 Datasets

We generated synthetic datasets mimicking the OA sources in different urban environments. These synthetic datasets mimic mass spectral OA analyses of a Time-of-Flight Aerosol Chemical Speciation Monitor (ToF-ACSM, Fröhlich et al. (2013)), a ubiquitous instrument in measuring PM composition with a focus on OA. The instrument measures a signal for several mass-to-charge ratio channels. We model these mass spectra as a sum of 2–5 different mixed sources. The sources are constructed

of time-independent chemical fingerprints, in our notation **F**, from the AMS Spectral Database (Ulbrich et al., 2009, 2022) combined with their time behaviour and magnitude, **G**. The noiseless spectra, **Z**, is then acquired by matrix multiplication of

**F** and **G**. We then generate **X**, as Equation 2, by applying random Gaussian noise to each data point. The errors are applied to **X**, which is a sum of all the components, so individual component error in the original **G** is undefined.

The ToF-ACSM alternates between measuring particles and air together, called open signal ($I_{open}$), and measuring only air, called closed signal ($I_{closed}$). The difference signal ($I_{diff}=I_{open}$-$I_{closed}$) represents the signal caused by the measured particles. For computing the error of $I_{diff}$, we use an error function based on the signal strength, according to Allan et al.

(2003), and Ulbrich et al. (2009):

$$Error_{open} = \sqrt{\frac{(I_{open} + I_{baseline}) \times t_{open}}{\sqrt{\frac{28}{m/z}}}} \tag{8}$$

$$Error_{closed} = \sqrt{\frac{(I_{closed} + I_{baseline}) \times t_{closed}}{\sqrt{\frac{28}{m/z}}}} \tag{9}$$

$$Error = max(Error_{min}, \frac{1.2 \times \sqrt{Error_{open}^2 + Error_{closed}^2}}{t_{open} \times \sqrt{\frac{28}{m/z}}}) \tag{10}$$

Where $t_{open}$ and $t_{closed}$ are the open and closed signal measurement times, respectively, $m/z$ is the mass charge ratio of

the measurement, $Error_{min}$ is a lower limit set on the measurement error, and $I_{baseline}$ is the baseline signal in the mass spectrometer. Since the organic fragments ions at the m/z values 16,17,18 and 28 are computed based on the measurement at m/z 44 and thus contain duplicate information, they are removed before running any of the models and only later reintroduced in the results.

## 3.1 Synthetic data representing a polluted megacity

First, we generated a synthetic ToF-ACSM OA mass spectral dataset mimicking a polluted megacity environment affected by multiple OA sources. The synthetic datasets used here are based on observations from Beijing, as it is a relatively well-studied environment. In our case, the modelled sources are traffic exhaust, HOA; cooking, COA; biomass burning, BBOA; coal combustion, CCOA; and secondary OA, OOA. In addition, we also constructed more simple datasets generated with fewer factors (2 factors: HOA + OOA, three factors: HOA+COA+OOA, and four factors: HOA+COA+BBOA+OOA). **F** were

chemical fingerprints from literature (Elser et al., 2016; Ulbrich et al., 2009, 2022). **G** was created as a mix of Gaussian and Cauchy (BBOA Gaussian, others Cauchy), biased, positive random walks, with added typical diurnal concentration cycles (Kulmala et al., 2021) corresponding to the matching OA sources (based on **F**). Mix of different random walk distributions was chosen to test if the model approximation of Cauchy auto-correlation works with them. The random walk aims to introduce variability such as one would get from varying transport and mixing. The diurnal cycle was simply summed to the random walk

to produce the time series. The components' (OA sources') overall order of magnitudes and diurnal concentration variability were estimated based on previous literature on OA sources in Beijing (Kulmala et al., 2021). For each number of factors, we

constructed ten different datasets amounting to a total of 40 datasets. For reference, one 5-component dataset in its component form is shown in Figure 2. CCOA and BBOA are very similar in $\mathbf{F}$ and $\mathbf{G}$ (See Appendix B), making it a challenging dataset. The time series of CCOA and BBOA are also similar in magnitude and have similar diurnal behaviour. While the short-term auto-correlation is high, the random walks of the megacity dataset are not as highly auto-correlated as the PM data in Figure 1. The added diurnal peaks can be seen as the periodic peaks in the correlograms in Figure 2c and f. See Appendix C for an example of the m/z dependence of the measurement errors on this dataset.

## 3.2 Synthetic dataset representing a typical European urban environment

As another test, we used chemical transport model data from Jiang et al. (2019) representing approximately two weeks of simulated measurements in Zurich, Switzerland. Zurich represents a typical European city with low pollution levels. In this case, the $\mathbf{G}$ time-series come from the transport model and $\mathbf{F}$ is taken from the literature (HOA and BBOA from an ambient analysis presented by (Elser et al., 2016; Ulbrich et al., 2009, 2022), biological SOA ($SOA_{bio}$) from an ambient analysis presented by Daellenbach et al. (2017), anthropogenic SOA, $SOA_{anthro}$, is represented by laboratory Diesel generator SOA presented by Sage et al. (2008)).

This dataset differs from those in Section 3.1 in two important ways. Firstly the concentrations are lower since the environment is less polluted, which affects the error estimation as larger relative measurement errors, median 1.0 % of data magnitude for this dataset and median 0.6% for one of the datasets described in Section 3.1. Secondly, the sources exhibit high correlation in the time series (Appendix B shows that correlations in $\mathbf{G}$ are higher than in the dataset described in Section 3.1), possibly indicating that meteorological conditions and transport of pollutants are important drivers of the concentration of the components. From a source apportionment analysis perspective, this simulates the worst-case situation with the data having poor separability in $\mathbf{G}$. The components of this dataset compared to the megacity data can be seen in Figure 2. The auto-correlation behaviour of the two datasets is very similar, indicating that our fully synthetic data behave as realistically as the transport model.

## 4 Results and discussion

In this section we compare the factorizations from BAMF and PMF on simulated megacity data, in Section 4.1, and synthetic European data, in Section 4.2. We also investigate what happens when we do not know the true number of sources, in Section 4.3, and how additional prior information improves the factorization, in Section 4.4.

### 4.1 Simulated megacity source apportionment

In the first experiment, we assess the performance of BAMF, BAMF-0, and PMF on synthetic data mimicking the conditions in a polluted megacity described in Section 3.1. First, we assess the reconstruction of the input by the different models. In addition to minimizing the residuals, BAMF also includes a penalty for deviations from auto-correlation in $\mathbf{G}$. Due to this and BAMF being a sampled model instead of an optimizer as described in Section 2.3, PMF would be expected to give answers

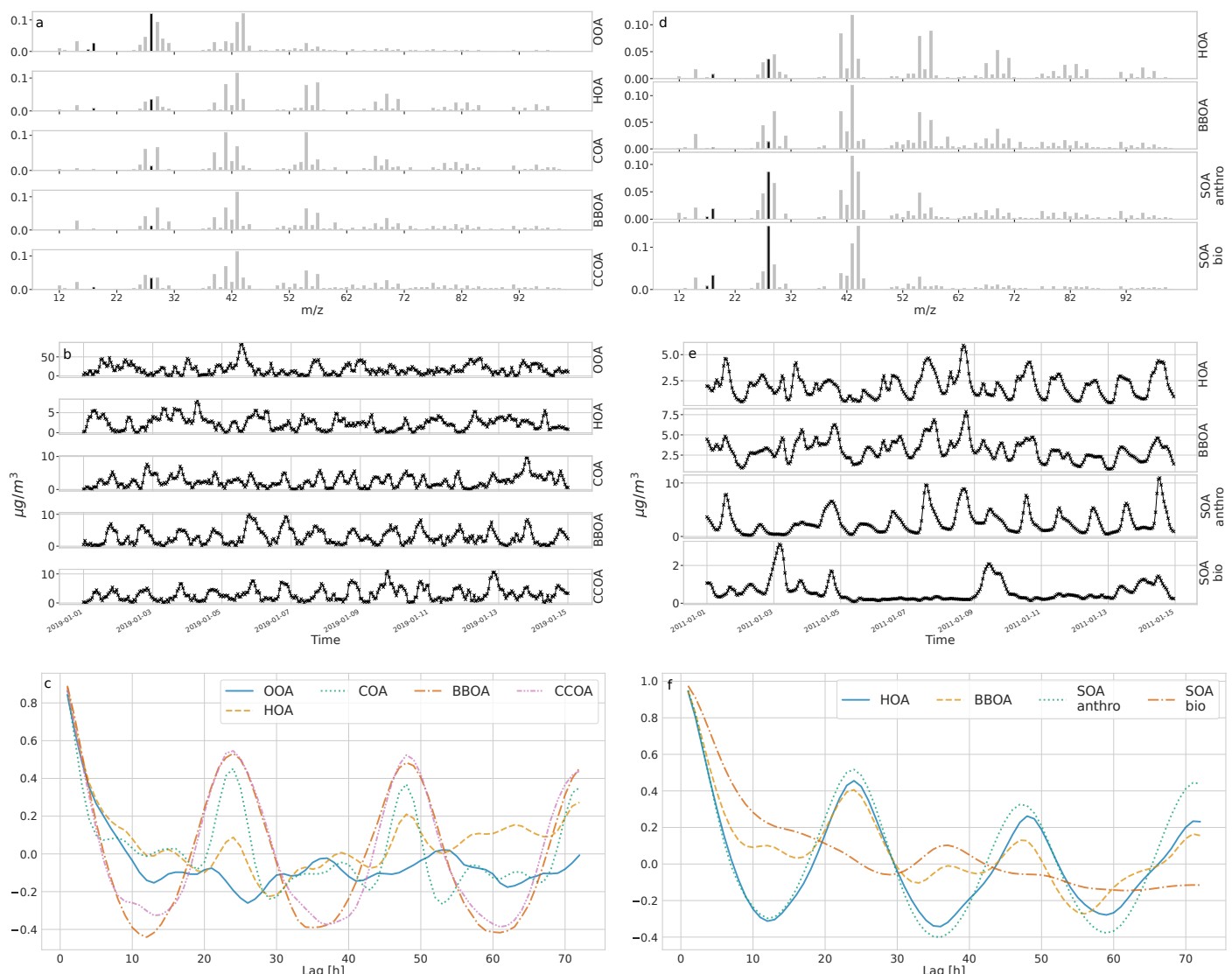

**Figure 2.** Characteristics of the synthetic ToF-ACSM OA datasets. Panels a and d show the factor profiles (**F**) used to construct the synthetic datasets. The solid black bars are ions derived from m/z 44 and are only used for converting concentrations. Panels b and e show the factor time series (**G**) used to construct each dataset, and the unit for them is $\mu g/m^3$, and panels c and f show each factor's temporal auto-correlation. Auto-correlation refers to the Pearson correlation coefficient of the component with the same component time-shifted by the number of hours. Panels a,b and c are for megacity data and panels d, e, and f are for the European urban environment.

with lower absolute measurement error weighted residuals compared to BAMF. In other words, PMF is expected to have a better reconstruction performance.

Figure 3 shows the median solution reconstruction across the ten different datasets as the number of components increases. All models reconstruct the input data well within the error estimate. The similar reconstruction metrics for BAMF and BAMF-0 suggest that the inclusion of the auto-correlation term does not substantially deteriorate the reconstruction accuracy. On the other hand, PMF has, as expected, marginally lower absolute measurement error-weighted residuals than BAMF and BAMF-0. However, they are below unity and within the error estimate, judging by the normalized residuals. Therefore, PMF likely finds solutions that fit the noise in the data better. All other reconstruction metrics ($\mathbf{S}$ mean, $\mathbf{S}$ standard deviation, $Q_m$*) are comparable for all models.

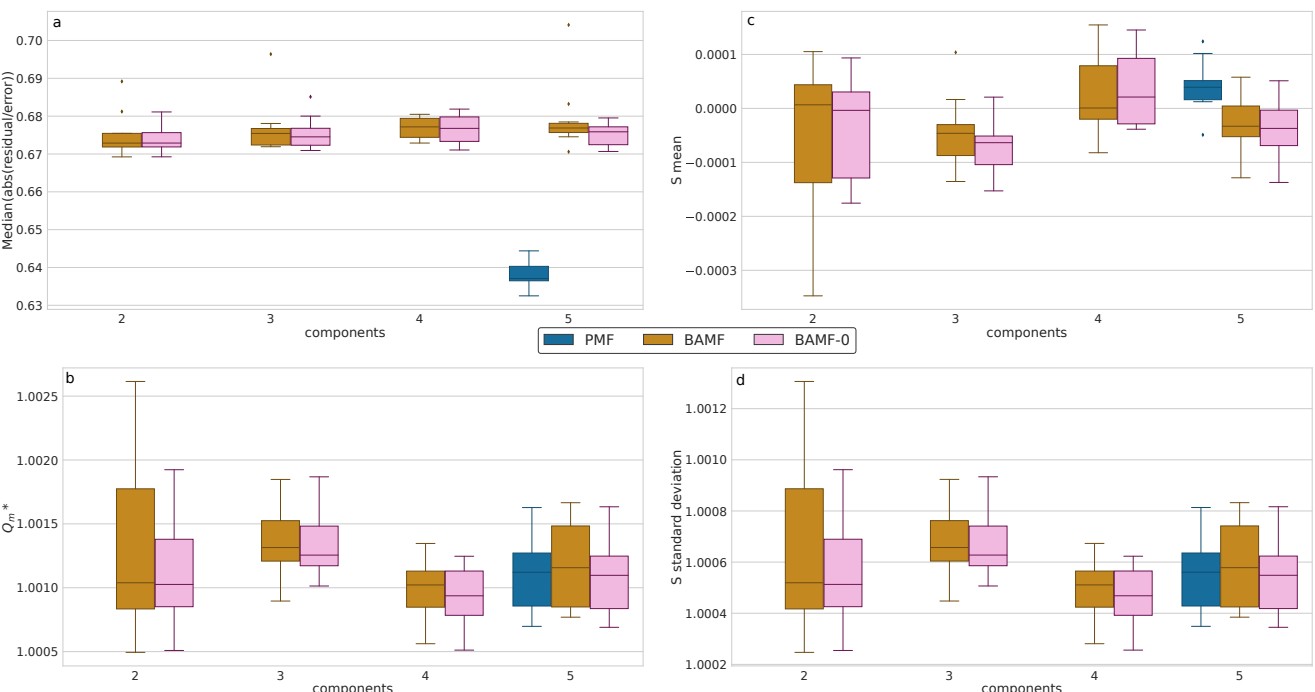

**Figure 3.** Reconstruction metrics for BAMF, BAMF-0, and PMF for synthetic megacity data. Panel a shows the relative error of $\mathbf{X}$ of the median solution as a function of the number of components for the synthetic megacity data (BAMF and BAMF-0: results from 10 different datasets for each number of factors, PMF: only for the 5-factor cases). Panel b shows the $Q_m*$ statistic, where all models are very similar. Panel c shows the mean and Panel d shows the standard deviation of $\mathbf{S}$, as a function of the number of factors for the synthetic megacity data (the ideal value for the mean is 0 and the ideal value for standard deviation is smaller than 1). $\mathbf{S}$ refers to the difference between the original data and the samples normalized with the standard deviation of the samples (uncertainty of the model).

Data reconstruction is essential to get within the error limits. However, source apportionment aims to accurately and precisely resolve the actual components in $\mathbf{G}$ and $\mathbf{F}$, i.e. factorization performance. Each model's solution to the example dataset is shown

in Figure 4. For this example, we observe that all models resolve all five components. However, BAMF has a better factorization performance both in $\mathbf{F}$ & $\mathbf{G}$ than the models not accounting for auto-correlation (BAMF-0, PMF) in Table 1. Similarly, for the diurnal cycles in the example in Figure 4c, the OA sources' diurnal concentration, as identified by BAMF, closely resembles the ground truth components. While all models capture the time behaviour of the diurnals, the absolute magnitude has a bias, with BAMF having a substantially smaller bias than the other models for four out of five components (Table 1).

All models slightly underestimate OOA, which results in overestimating the other components (Table 1). For the final component, CCOA, PMF has less bias, but it correlates worse with the truth. BAMF-0 and PMF significantly underestimate OOA, which results in an overestimation of other components to get the reconstruction correct. Some factors have a more pronounced cyclical temporal behaviour as witnessed by high auto-correlation at higher temporal lag (e.g., Figure 4c CCOA, BBOA, COA). Based on Table 1, some of the factors with pronounced cyclical temporal behaviour are better represented by BAMF than PMF (e.g., BBOA, COA), while this is not necessarily the case for others (e.g., CCOA).

When considering all ten synthetic datasets with five components mimicking a polluted megacity, BAMF consistently produces factors closer in magnitude to the truth and which correlate better with the actual factors than the other models (Figure 5). BAMF is also better correlated with the time behaviour of the components, while one of the components (CCOA, strongly correlated with BBOA in terms of $\mathbf{G}$ and $\mathbf{F}$) is difficult for all models. We hypothesize BAMF-0 shows better spread Figure 5, due to being constrained by the underestimation of OOA and having to include those peaks in the spectra. Figure 5a, shows that BAMF can over- and underestimate both BBOA and CCOA depending on the dataset. Appendix A shows the inverse relationship between CCOA and BBOA mass concentration biases and how the profile reconstruction affects the CCOA mass concentration bias for the BAMF model. Overall, using auto-correlation in source apportionment markedly improves the quality of the resolved factors while keeping the overall reconstruction metrics similar.

## 4.2 Simulated European low pollution city source apportionment

In a second exercise, we assessed the performance of BAMF, BAMF-0, and PMF on a synthetic dataset mimicking the conditions in a typical European city (Section 3.2). In contrast to the fully synthetic dataset in Section 3.1, here, the true components $\mathbf{G}$ are OA source components computed by an air quality model (Jiang et al., 2019). This provides $\mathbf{G}$ time series close to the atmosphere while still knowing the ground truth. While the three models show somewhat different components (Figure 6), the reconstruction metrics indicate that all models have acceptable solutions (Table 2). In fact, the metrics also show that the European dataset is reconstructed almost within the error limits with already only three components, i.e. 1 component less than is present in the synthetic dataset (HOA, BBOA, $SOA_{anthro}$, $SOA_{bio}$). This could explain why there is significant freedom in acceptable 4-component solutions and variation between them.

All models show signs of mixing between the components, likely due to the correlation of the true $\mathbf{G}$ components (time behaviour is very similar) as well as similar chemical signatures $\mathbf{F}$. PMF mixes the SOA components while BAMF mixes the POA components. However, it is worth noting that BAMF has a significant bias on several components in $\mathbf{G}$ as seen in Table 2, but otherwise reflects their time behaviour well. Given the recovered POA/SOA ratio, BAMF likely mixes HOA and BBOA explaining that HOA is overestimated while BBOA is underestimated. PMF, on the other hand, produces two almost identical

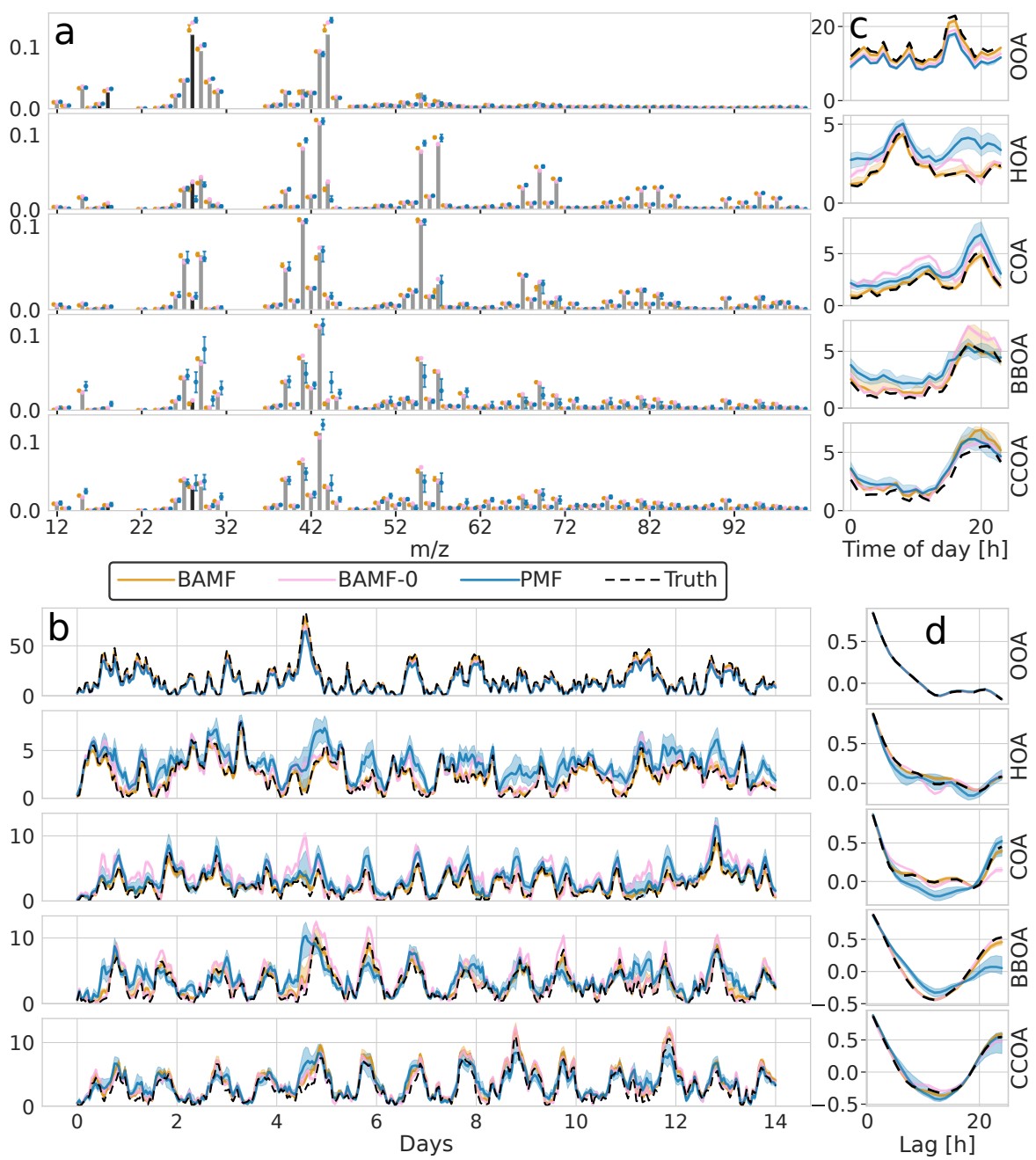

**Figure 4.** Illustration of **F** and **G** reconstruction of all models for one of the synthetic megacity ToF-ACSM OA datasets with five components. Panel a) is **F**, b) is **G**, c) is the diurnal concentration variation and d) is auto-correlation measured as Pearson correlation coefficient. Here, we display the median and the 0.25 and 0.75 quantiles. For all BAMF-type models, these quantities are computed based on the samples and for PMF, based on the 100 solutions.

**Table 1.** Reconstruction and factorization performance of all three models for the synthetic megacity ToF-ACSM OA dataset in Figure 4. The reconstruction metrics measure the residuals divided by the error estimate. A value closer to 0 is better and a value below 1 is smaller than the error estimate given to the model. The factorization performance is assessed via three metrics: **G** / Truth is the average ratio of each factor time series, r is the Pearson correlation coefficient between the factor time series, and $\rho$ is the Spearman correlation coefficient between the factor profiles, For the factorization performance, a value closer to 1 is better. Nonlinear correlation coefficient is used for factor profiles, since they have an additional constraint of summing to unity and thus linear correlation is penalized for small errors disproportionately. For each metric, the best value is highlighted in bold.

| | | BAMF | BAMF-0 | PMF |
|---|---|---|---|---|
| Reconstruction performance: | Median($\|Z - X\|/\sigma$) | 0.68 | 0.68 | **0.64** |
| | Max($\|Z - X\|/\sigma$) | 5.85 | 5.07 | **3.72** |
| Factorization performance: | **G** / Truth OOA | **0.94** | 0.83 | 0.78 |
| | **G** r OOA | **1.00** | **1.00** | **1.00** |
| | **F** $\rho$ OOA | **0.99** | 0.90 | 0.93 |
| | diurnal **G** / Truth OOA | **0.94** | 0.83 | 0.78 |
| | **G** / Truth HOA | **0.99** | 1.16 | 1.51 |
| | **G** r HOA | **0.99** | 0.92 | 0.86 |
| | **F** $\rho$ HOA | 0.99 | **1.00** | 0.97 |
| | diurnal **G** / Truth HOA | **1.01** | 1.21 | 1.57 |
| | **G** / Truth COA | **0.99** | 1.44 | 1.39 |
| | **G** r COA | **0.98** | 0.78 | 0.95 |
| | **F** $\rho$ COA | 0.99 | **1.00** | **1.00** |
| | diurnal **G** / Truth COA | **1.03** | 1.53 | 1.47 |
| | **G** / Truth BBOA | **1.08** | 1.26 | 1.28 |
| | **G** r BBOA | 0.98 | **1.00** | 0.66 |
| | **F** $\rho$ BBOA | 0.99 | **1.00** | 0.94 |
| | diurnal **G** / Truth BBOA | **1.10** | 1.26 | 1.37 |
| | **G** / Truth CCOA | 1.24 | 1.20 | **1.17** |
| | **G** r CCOA | **0.99** | 0.98 | 0.86 |
| | **F** $\rho$ CCOA | **1.00** | 0.99 | 0.96 |
| | diurnal **G** / Truth CCOA | 1.27 | **1.19** | 1.27 |

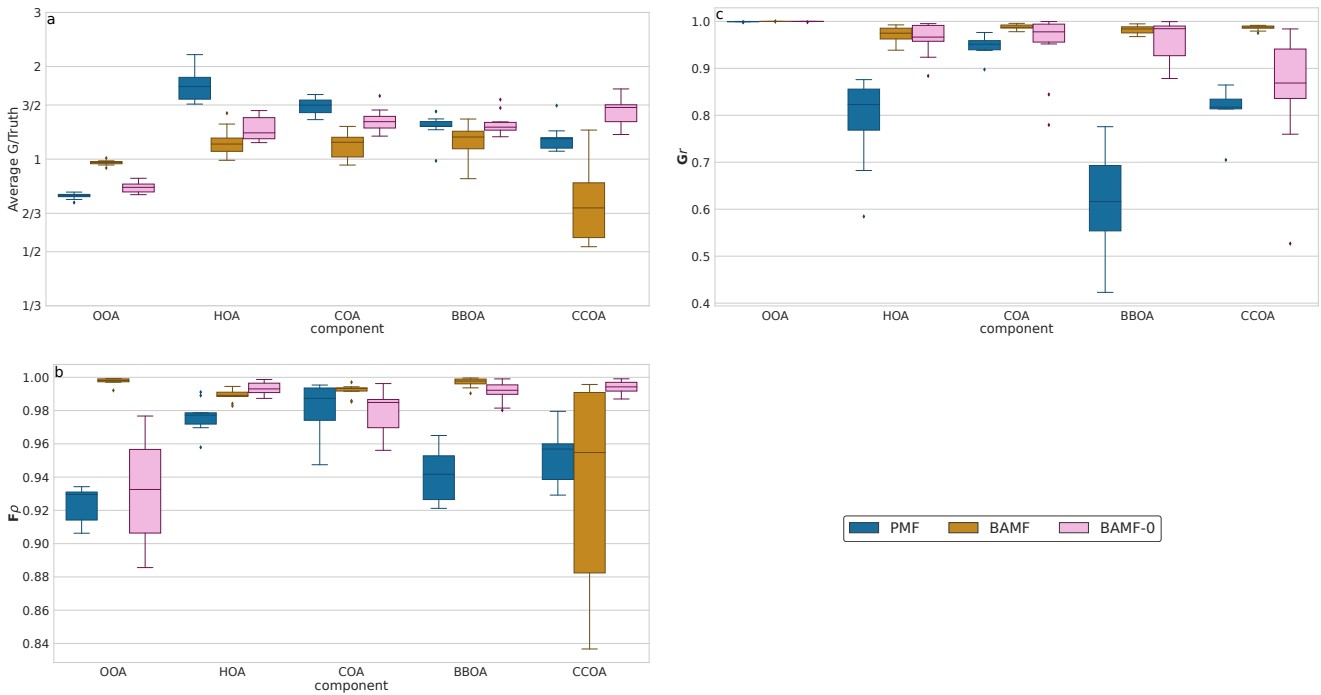

**Figure 5.** Summary of factorization performance of the three models for all synthetic megacity ToF-ACSM OA datasets with five components (10 datasets): Panel a shows the mean of the median value of the components of **G** divided by the true value (ideal value is 1). Panel b shows the Spearman correlation coefficient median solution components of **F** compared to the true value (a value of 1 refers to a perfect correlation). Panel c shows the Pearson correlation coefficient of the median solution components of **G** compared to the true value (a value of 1 refers to a perfect linear correlation).

components (both in **F** and **G**) for two of the four components, and PMF cannot thus distinguish the components present in the dataset. In Figure 6d BAMF-0 and BAMF also seem to overestimate higher lag auto-correlation of BBOA in a similar fashion but have a higher bias. It should be noted that even BAMF only considers lag-1 auto-correlation in the model. For HOA, BAMF and PMF don't match the anti-correlated part between lags 5 to 20 and PMF can't match the behaviour of SOA bio. Overall, all models are challenged by the European dataset, with BAMF having the most consistent performance.

## 4.3 Resolving an unknown amount of sources

For real-world source apportionment analyses, the true amount of components, i.e. sources, to be resolved via matrix factorization is unknown yet crucial. Despite the importance, accurately determining and specifying the correct number of modelled components is not trivial, see, e.g. Isokääntä et al. (2020); Ulbrich et al. (2009); Zhang et al. (2011). Typical strategies rely on reconstructing **X** within the measurement error, the absence of structure in the measurement error weighted residuals and the resolved components' environmental interpretability. Here, we assess the behaviour of the BAMF, BAMF-0, and PMF

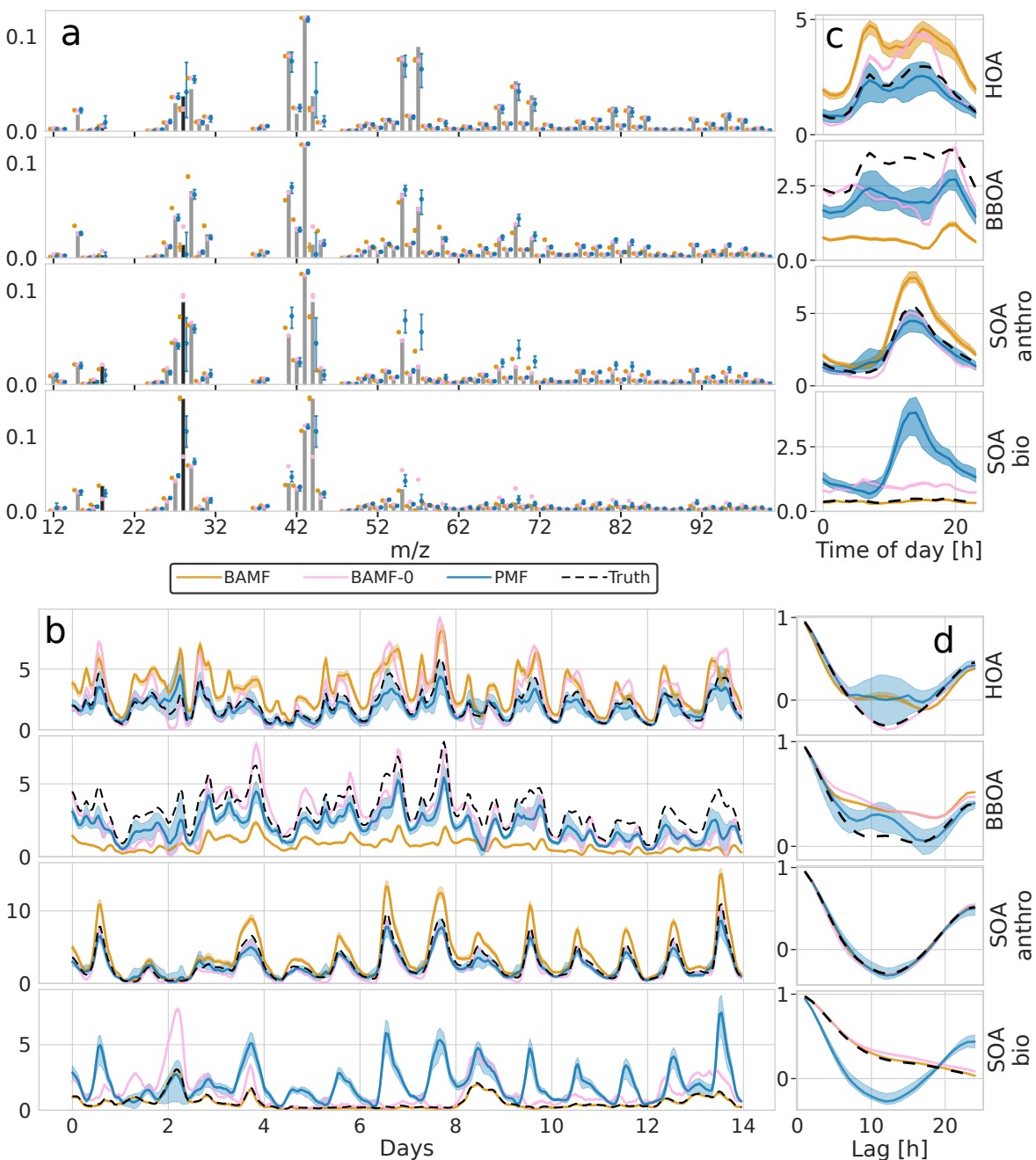

**Figure 6.** Factorization performance of all three models for the synthetic European city ToF-ACSM OA dataset. The shaded area is the interquartile range (0.25 to 0.75 quantile). Panel a) is **F**, b) is **G**, c) is the diurnal and d) is auto-correlation measured as Pearson correlation coefficient.

models as the number of components are changed on the 4-component chemical transport model dataset from Section 3.2. The model runs were performed both with an underspecified setting using three components and overspecified setting using five components (Figures 7 and 8). While underspecified, all models extract an $SOA_{anthro}$ component and merge the remaining three components into two. At the same time, BAMF extracts a component similar to $SOA_{bio}$, BAMF-0 and PMF extract components similar to HOA and BBOA but lack $SOA_{bio}$.

For the overspecified models (5 instead of 4 components), the results differ (Figure 8). While the models without the auto-correlation assumption (PMF, BAMF-0) split the true components into multiple sub-components (mostly the POA components, HOA and BBOA), BAMF produces an extra component that is easily identifiable as unnecessary in addition to the four components resolved with four factors. This unnecessary component is characterized by an extremely high auto-correlation and low magnitude, as seen in Figure 8b, c, and d. For this component the time series is almost constant, and the composition is flat with large uncertainties. The extra component doesn't affect the factorization performance of BAMF's other components substantially. At the same time, BAMF-0 and PMF have a reduced factorization performance with too many components (Figure 8, Table 2). In general use, one would prefer the model to indicate the limits of the factorization as BAMF does, instead of producing duplicate components. It should be noted that the behaviour of BAMF can be changed by adding new model terms, such as constraints on $\mathbf{F}$. For example, the model minimizes the constrained component when it is run equally overspecified (5 instead of 4 components) but with a priori information on $\mathbf{F}$ as can be seen in Appendix D.

## 4.4 Using ancillary information to improve resolving sources

As highlighted above in Section 4.2, all models are challenged by the European dataset. Imperfect matrix factorization results are likewise often observed when using PMF for real-world chemical datasets, see, e.g., Canonaco et al. (2013); Daellenbach et al. (2017). Often, information is available that could help resolve the sources, such as chemical fingerprints of specific components associated with different sources. In current practice, previously observed $\mathbf{F}$ profiles are often used as boundary conditions in source apportionment analyses. This approach has significant uncertainty in the general case because true $\mathbf{F}$ is unknown. Still, in our specific test case, the a priori information is precisely correct—the known information on $\mathbf{F}$ of the true components.

We tested the models' performance when using a priori information on $\mathbf{F}$ using three different approaches on the European dataset (from Section 3.2):

1. Full constraint: for BAMF-C and PMF, the two POA components (HOA & BBOA) were fully (for all m/zs) constrained with a roughly 18% allowed deviation from the anchor (see Section 2.2.2 for the determination of this).

2. Incomplete constraint: Knowledge on the entire factor profile is not always available, e.g., not the same m/z range is measured. With the incomplete constraint approach, we test whether a priori information on parts of the factor profile improves the factorization performance and thus whether such information is useful. For the BAMF model, a priori information was only used in a limited arbitrarily chosen m/z range (12–60) instead of for all m/zs (m/z 12–100), same allowed deviation from the anchor for the constrained components.

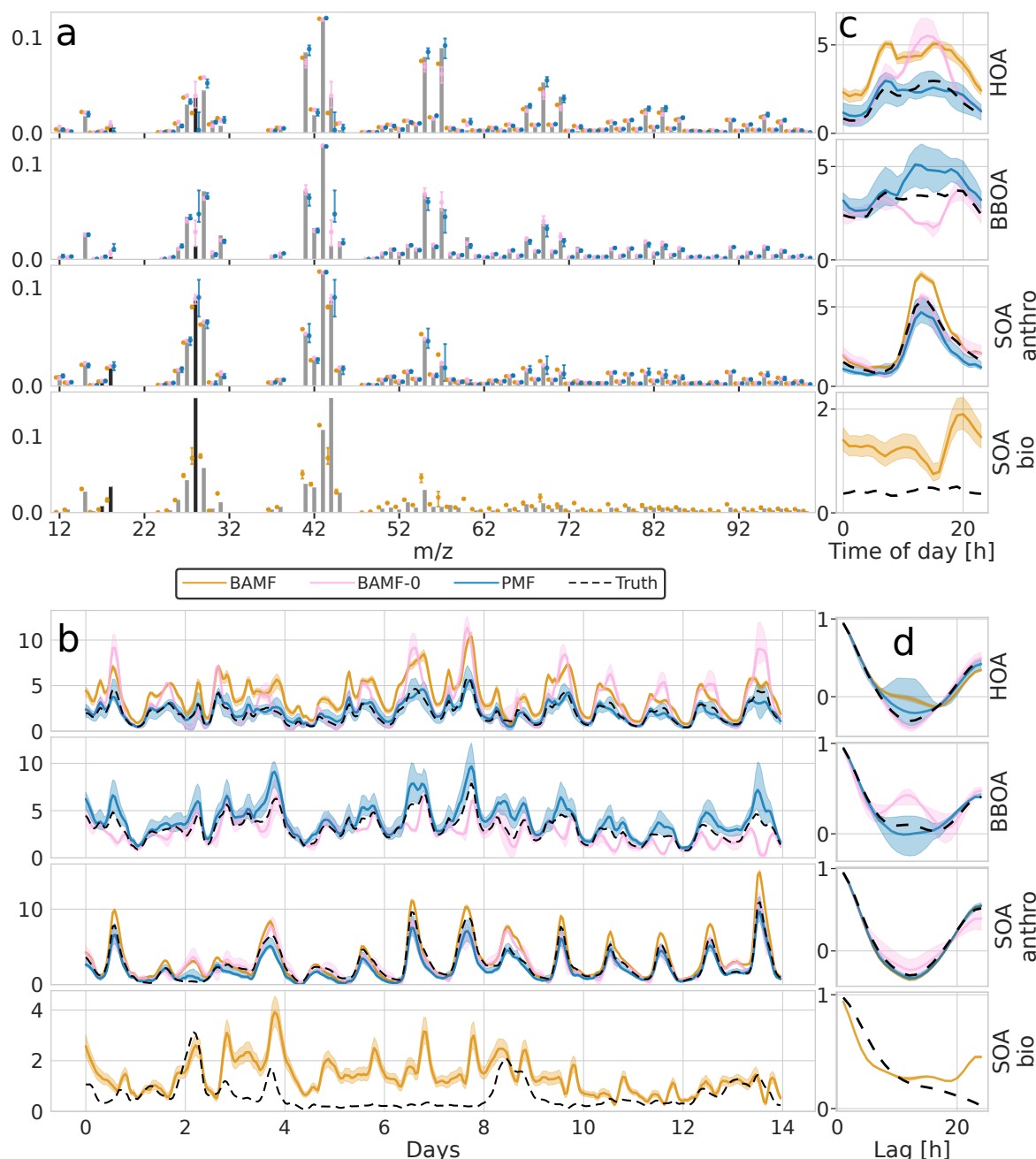

**Figure 7.** Factorization performance of underspecified (3 components) models for the synthetic European city ToF-ACSM OA dataset. Panel a) is **F**, b) is **G**, c) is the diurnal and d) is auto-correlation measured as Pearson correlation coefficient. The models extract different components and they are shown next to the closest original component. This is why there are 4 components shown even though the models extract only 3 components each.

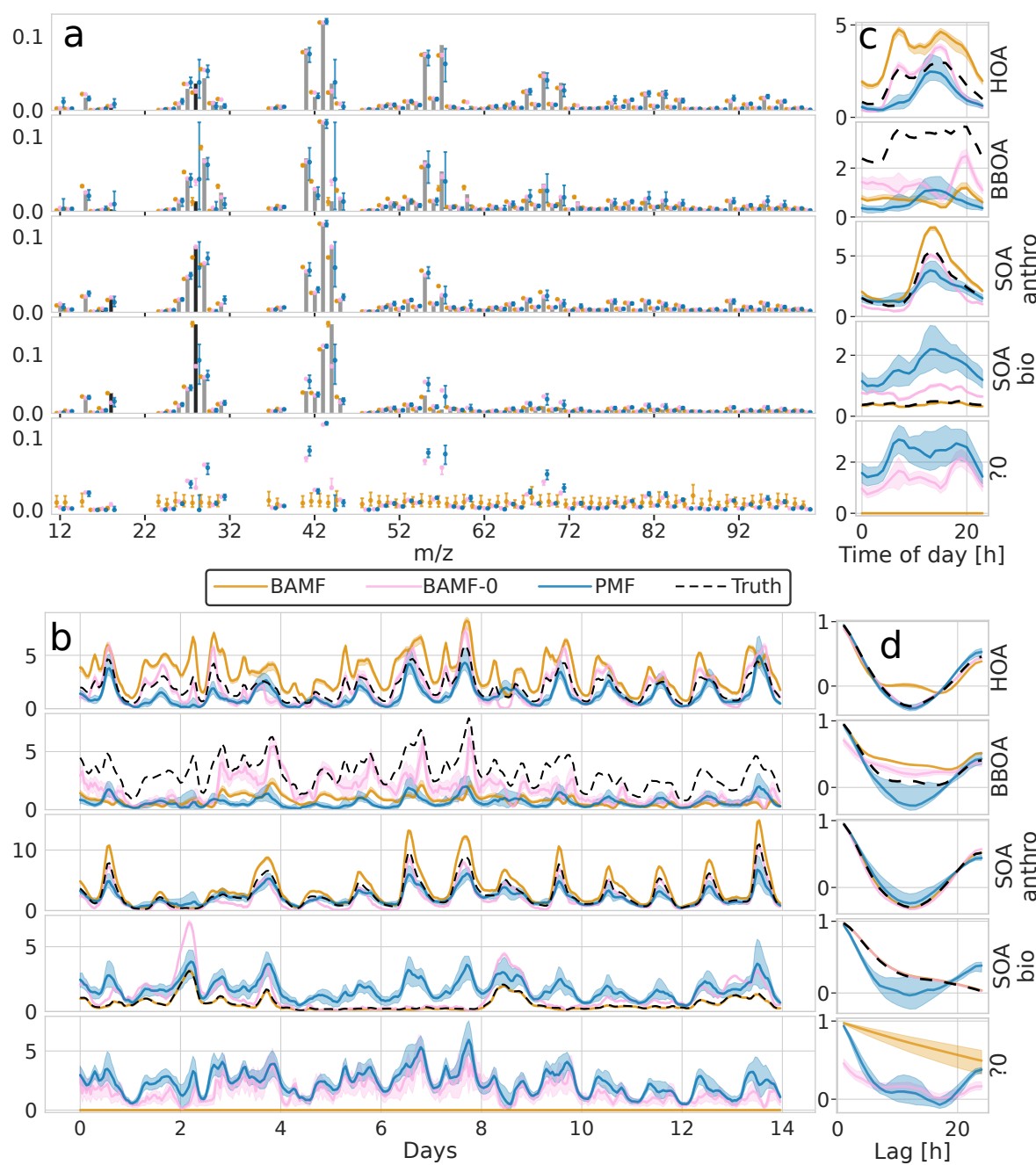

**Figure 8.** Factorization performance of overspecified (5 components) models for the synthetic European city ToF-ACSM OA dataset. Panel a) is **F**, b) is **G**, c) is the diurnal and d) is auto-correlation measured as Pearson correlation coefficient. "?0" denotes an unidentified component.

3. Partial constraint: Sometimes, only very little chemical information is available for a specific factor, and it is defined by few key tracers. With the partial constraint, we test whether a priori information on just a few peaks/variables improves the factorization performance. For the BAMF model, a priori information was only used for 4 arbitrarily chosen m/z peaks out of 74 (m/zs 45, 57 and 60, F[ii,jj], compared to m/z 43, F[kk,ll]) for HOA & BBOA defined with the same allowed deviation from the anchor for constrained components.

The reconstruction and factorization performance of the different models are compared in Table 2. The fully constrained BAMF model (BAMF-C) performs substantially better in extracting BBOA both in $\mathbf{F}$ and $\mathbf{G}$ compared to BAMF. In fact, the extracted components are very similar to the true components with very similar temporal behaviour and reduced biases in $\mathbf{G}$ (Figure 9). Fully constrained PMF performs marginally better than unconstrained PMF but still mixes $SOA_{bio}$ with $SOA_{anthro}$, Figure 9. This can be expected since the a priori information is applied to HOA and BBOA, not to the SOA components. Figure 9d shows that PMF-C captures HOA time behaviour very well, while BAMF still overestimates some auto-correlation, even though the bias in Figure 9a is significantly reduced. PMF-C also has some solutions that start to approach correct SOA bio, with solutions falling between unconstrained PMF and BAMF-C. For BAMF-C and BAMF SOA bio is virtually unchanged (Figures 9a, 9d, 6a and 6d). The incompletely constrained BAMF model performs slightly worse than the fully constrained BAMF model. Partially constrained BAMF performs worse but is still on par with the fully constrained PMF (Table 2). The partially constrained BAMF model reduces bias in BBOA but starts mixing SOA bio with the other components (Figure 10, Table 2). Using a priori information on $\mathbf{F}$ improves the factorization performance of both PMF and especially BAMF, with more information leading to solutions closer to the ground truth. This is especially helpful when the additional information is on the components the model mixes when unconstrained. Such prior information is powerful, for example constrained PMF does better than unconstrained BAMF for all components except SOA bio. For this dataset and these constraints, BAMF-C fully constrained has the best factorization performance, fully constrained PMF and partially constrained BAMF-C are performing slightly worse but about equally good and unconstrained models fail to resolve one or more components correctly regardless of model. Comparison of the results of fully constrained PMF and unconstrained BAMF can be seen in Appendix G.

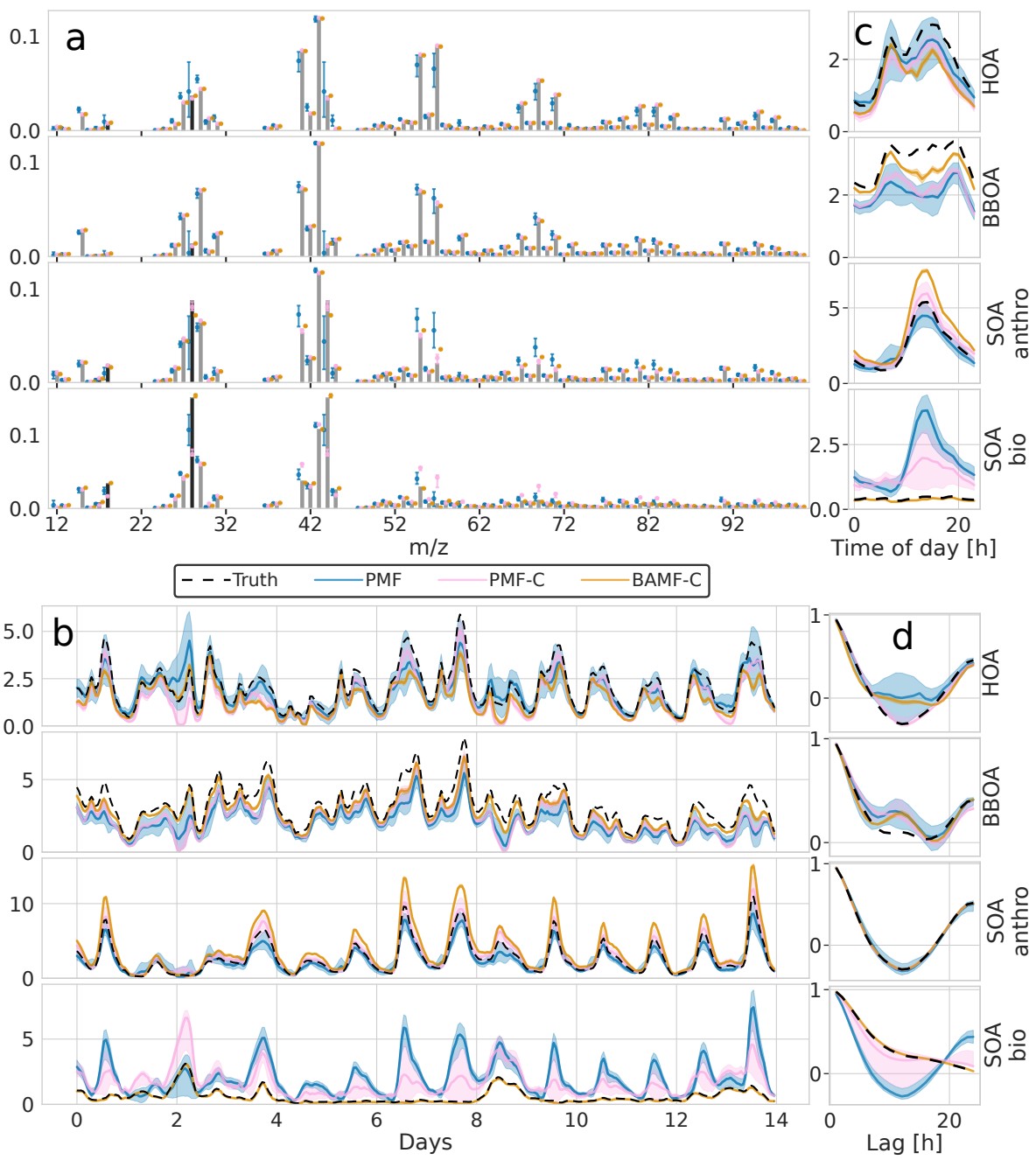

**Figure 9.** Factorization performance of models using a priori information on **F** for the synthetic European city ToF-ACSM OA dataset, fully constrained BAMF and fully constrained PMF results compared to unconstrained PMF. Panel a is **F**, b is **G**, c is the diurnal concentration, and d is the auto-correlation behaviour.

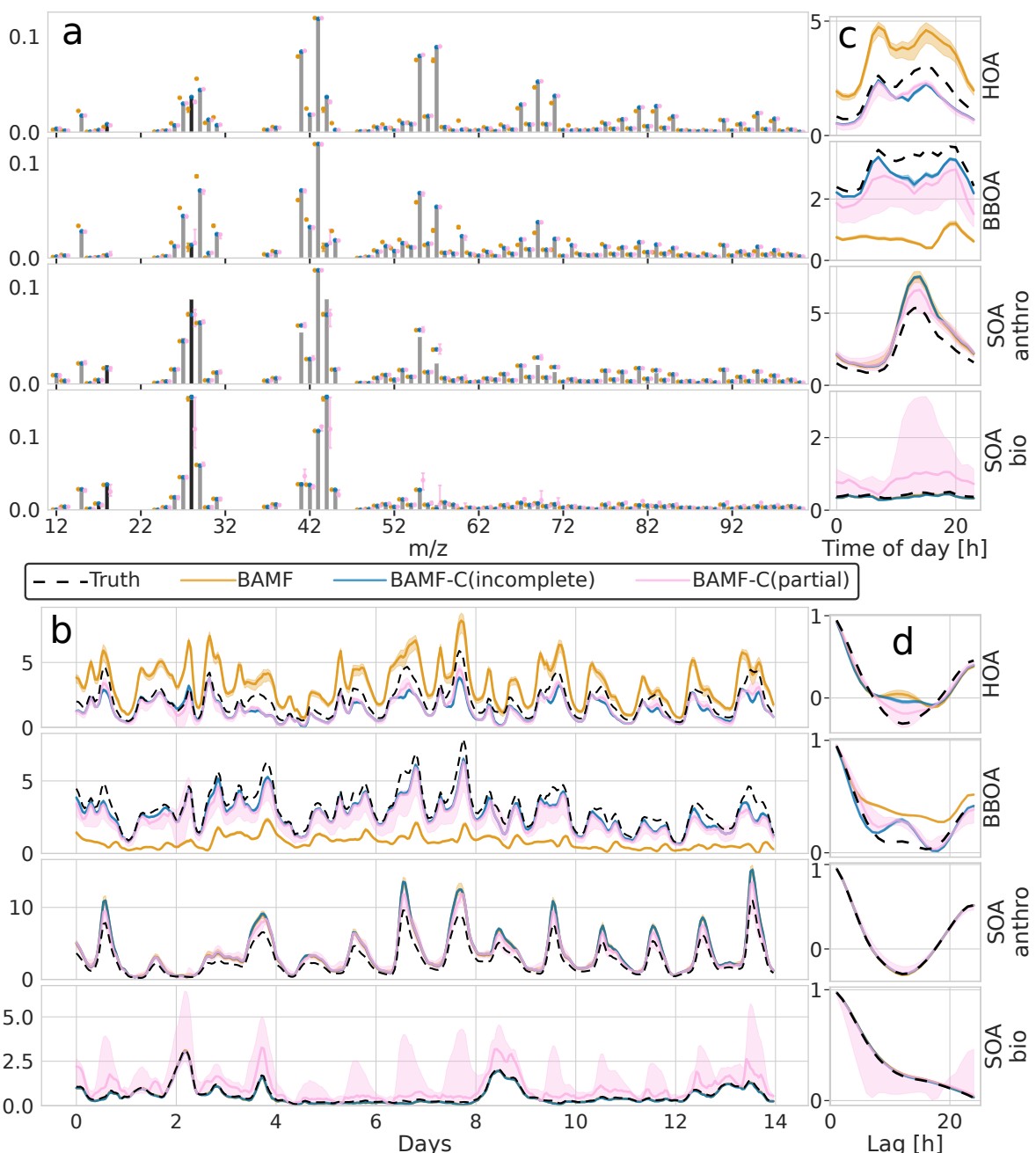

**Figure 10.** Factorization performance of models using a priori information on **F** for the synthetic European city ToF-ACSM OA dataset, partial and incomplete constraints in BAMF compared to unconstrained BAMF. Panel a is **F**, b is **G**, c is the diurnal concentration and d is the auto-correlation behaviour. The results for SOA components overlap between BAMF and BAMF-C(incomplete) such that the BAMF results are not visible in panels b, c, and d.

**Table 2.** Reconstruction and factorization performance for the synthetic European city ToF-ACSM OA dataset. **G / True** is the average ratio of the component to the true one, r is the Pearson correlation coefficient, and $\rho$ is the Spearman correlation coefficient. Diurn is a factor's average ratio of the diurnal of G to the true diurnal. POA / SOA is the ratio of primary (HOA, BBOA) and secondary organic aerosol (anthropogenic SOA, biogenic SOA); for the ground truth, this ratio is 1.55. Unidentified components were not included in this ratio. For the reconstruction metrics, values closer to 0 are better, and values below 1 are within the error given to the model. For factorization metrics, the ideal value is 1.

| | | | $\|X-Z\|/\sigma$ | | HOA | | | | SOA anthro | | | | SOA bio | | | | BBOA | | | | POA |
|---|---|---|---|---|---|---|---|---|---|---|---|---|---|---|---|---|---|---|---|---|---|
| | | | Median | Max | G/True | G r | F $\rho$ | Diurn | G/True | G r | F $\rho$ | Diurn | G/True | G r | F $\rho$ | Diurn | G/True | G r | F $\rho$ | Diurn | / SOA |
| BAMF | Full | 3 | 1.18 | 54.56 | 0.78 | 0.98 | 0.97 | 0.80 | 1.55 | 0.95 | 0.99 | 1.53 | | | | | 0.85 | 0.97 | 0.99 | 0.86 | 1.01 |
| | | 4 | 0.67 | 5.13 | 0.72 | 0.91 | 1.00 | 0.74 | 1.39 | 1.00 | 0.99 | 1.39 | 0.91 | 1.00 | 0.96 | 0.88 | 0.86 | 0.97 | 1.00 | 0.87 | 0.96 |
| | | 5 | 0.67 | 5.14 | 0.04 | 0.16 | 1.00 | 0.05 | 1.41 | 1.00 | 0.99 | 1.42 | 0.96 | 1.00 | 0.95 | 0.93 | 0.50 | 0.77 | 1.00 | 0.48 | 0.38 |
| | Incomplete | 4 | 0.67 | 5.06 | 0.71 | 0.91 | 1.00 | 0.73 | 1.41 | 1.00 | 0.98 | 1.41 | 0.90 | 1.00 | 0.96 | 0.87 | 0.85 | 0.97 | 1.00 | 0.87 | 0.95 |
| | Partial | 3 | 1.18 | 53.08 | 0.96 | 0.90 | 0.96 | 1.00 | 1.10 | 0.93 | 0.98 | 1.11 | | | | | 0.96 | 0.91 | 0.97 | 0.95 | 1.66 |
| | | 4 | 0.67 | 4.98 | 0.73 | 0.96 | 1.00 | 0.74 | 1.32 | 0.99 | 0.99 | 1.33 | 1.59 | 0.91 | 0.98 | 1.96 | 0.80 | 0.97 | 1.00 | 0.75 | 0.88 |
| | | 5 | 0.67 | 5.03 | 0.28 | 0.93 | 1.00 | 0.30 | 1.40 | 1.00 | 0.99 | 1.40 | 0.90 | 1.00 | 0.96 | 0.91 | 0.10 | 0.17 | 0.99 | 0.12 | 0.21 |
| | None | 3 | 1.18 | 52.80 | 1.91 | 0.91 | 0.95 | 2.00 | 1.21 | 0.96 | 0.99 | 1.22 | 2.24 | 0.35 | 0.88 | 3.12 | | | | | 0.83 |
| | | 4 | 0.67 | 4.96 | 1.70 | 0.91 | 0.95 | 1.77 | 1.39 | 1.00 | 0.99 | 1.39 | 0.91 | 1.00 | 0.96 | 0.89 | 0.24 | 0.74 | 0.95 | 0.24 | 0.97 |
| | | 5 | 0.67 | 5.07 | 1.70 | 0.91 | 0.95 | 1.77 | 1.36 | 1.00 | 0.99 | 1.36 | 0.92 | 1.00 | 0.96 | 0.91 | 0.24 | 0.75 | 0.96 | 0.23 | 0.99 |
| BAMF-0 | None | 3 | 1.18 | 52.60 | 1.48 | 0.96 | 0.95 | 1.48 | 1.01 | 0.93 | 0.99 | 1.01 | | | | | 0.85 | 0.78 | 0.98 | 0.86 | 2.07 |
| | | 4 | 0.67 | 5.17 | 1.23 | 0.97 | 0.95 | 1.22 | 0.82 | 0.99 | 1.00 | 0.83 | 2.24 | 0.98 | 0.91 | 2.25 | 0.77 | 0.74 | 0.98 | 0.72 | 1.37 |
| | | 5 | 0.67 | 5.24 | 0.96 | 0.96 | 0.96 | 0.97 | 0.76 | 0.98 | 1.00 | 0.78 | 1.95 | 1.00 | 0.92 | 1.90 | 0.53 | 0.70 | 0.97 | 0.44 | 1.11 |
| PMF | Full | 3 | 1.10 | 82.93 | 0.82 | 0.97 | 1.00 | 0.85 | 1.79 | 0.97 | 0.98 | 1.78 | | | | | 0.62 | 0.80 | 1.00 | 0.65 | 0.74 |
| | | 4 | 0.63 | 4.01 | 0.73 | 0.97 | 1.00 | 0.74 | 1.12 | 1.00 | 1.00 | 1.13 | 2.78 | 0.93 | 0.92 | 3.15 | 0.71 | 0.90 | 1.00 | 0.70 | 0.78 |
| | | 5 | 0.62 | 4.29 | 0.70 | 0.97 | 1.00 | 0.71 | 0.57 | 1.00 | 0.99 | 0.58 | 1.79 | 0.83 | 0.96 | 2.10 | 0.54 | 0.88 | 1.00 | 0.52 | 1.18 |
| | None | 3 | 1.12 | 84.37 | 1.05 | 0.93 | 0.89 | 1.06 | 0.78 | 0.98 | 0.99 | 0.80 | | | | | 1.28 | 0.94 | 0.97 | 1.25 | 2.88 |
| | | 4 | 0.63 | 4.01 | 0.90 | 0.92 | 0.95 | 0.88 | 0.89 | 0.98 | 0.95 | 0.93 | 3.29 | 0.33 | 0.93 | 4.33 | 0.65 | 0.93 | 0.97 | 0.66 | 0.87 |
| | | 5 | 0.62 | 4.47 | 0.62 | 0.84 | 0.95 | 0.61 | 0.82 | 0.98 | 0.98 | 0.85 | 2.81 | 0.63 | 0.94 | 3.76 | 0.21 | 0.68 | 0.88 | 0.20 | 0.48 |

## 5 Conclusions

We present a Bayesian matrix factorization model that accounts for components' temporal auto-correlation (BAMF) and provides direct error estimation. BAMF is built on top of Stan, a freely available, robust, actively developed, open-source framework for statistical modelling with the ability of full Bayesian statistical inference with Markov-Chain-Monte-Carlo sampling. Here, we characterize BAMF's performance on synthetic Time-of-Flight Aerosol Chemical Speciation Monitor mass spectral OA data compared to PMF. This approach allows for assessing the model's performance based on input data reconstruction and the ability to accurately model components' chemical composition and concentration time series.

All models performed well in reconstruction performance regardless of factorization performance, indicating that reconstructing the data is insufficient to judge how good the extracted factors are. Without strongly correlated components, BAMF resolves temporally auto-correlated components well (synthetic megacity dataset), while PMF performs considerably worse. Both BAMF and PMF are challenged by strongly correlated components (European data).

Further, we show that using a priori information on the components' chemical composition improves BAMF factorization performance such that all components are well represented. Even adding a priori information for a few peaks significantly

reduced component bias, and partially specifying the profile (for 56% of the peaks) produced comparable results to fully constraining the profile with PMF. This opens up possibilities for using incomplete chemical composition information to improve factorizations.

While we presently test BAMF on synthetic OA ToF-ACSM data, source apportionment analyses of other chemical PM data (e.g., trace elements from either Xact or offline filter analysis) could also profit from accounting for the auto-correlation of components, if the components are auto-correlated. Further testing is especially needed for datasets with temporally sparse sources, i.e., pollution sources occurring only during specific events, which are also challenging for PMF.

Overall, we believe BAMF-type models are promising tools for source apportionment and deserve further research, e.g., improving the separation of the chemical composition of components or the computational speed of BAMF. These models can also be used complementary to current source apportionment methods due to their different emphasis and advantages. One such research topic would be introducing rolling window methods as has been done with PMF, to allow the source profiles to change over time and to act as a basis for real-time source apportionment. Other possible topics are using BAMF with other time series instruments and with real world data. Another area of development is computational speed, for the dataset sizes discussed here running BAMF takes few hours on a modern computer (Intel Xeon Silver 4110), but the time increases as the data size increases.

*Code and data availability.* The datasets will be available upon publication. The code will be available under an open source license upon publication.

*Author contributions.* AR, AB, KD, and KP participated in the model, dataset, and experiment development. AR ran and analyzed the experiments. KD and MM contributed to the PMF model solutions. JJ did the transport model runs. All authors contributed to the writing of the manuscript.

*Competing interests.* The authors declare to have no conflicts of interests.

*Acknowledgements.* We thank the Research Council of Finland for its support (decisions 337549, 345704, and 346376). KRD acknowledges support by SNSF Ambizione grant PZPGP2_201992. JJ acknowledges support by Science and Technology Commission of Shanghai Municipality, China (Shanghai Pujiang Program, 21PJ1402800).

## Appendix A: The correlation of BAMF CCOA error in the five component megacity datasets

The error in F for CCOA seems to be correlated with BBOA error, Figure A1, with correlation coefficient -0.5 and the error quickly increases as the component is underestimated as shown in Figure A2. Thus it seems that the variance in the reconstruction for CCOA in BAMF is due to the mixing of BBOA and CCOA components. This is probably due to these components having very similar **G** and **F** profiles as seen in Tables B1 and B2.

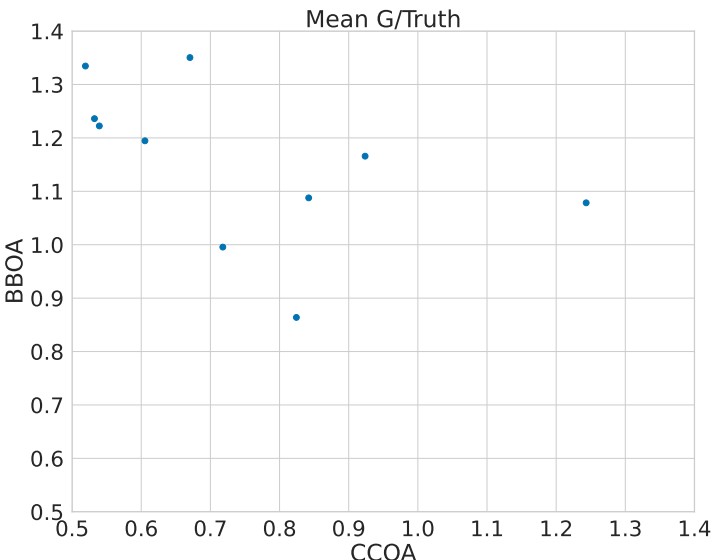

**Figure A1.** The bias of CCOA and BBOA **G** component in the 5 component megacity datasets.

## Appendix B: Correlation between true components in the datasets

Comparing the true components of the datasets shows that the European dataset components are much more correlated both in **G** Tables B1 and B3, as well as **F** Tables B2 and B4. Overall **F** components have similar, very high, correlations with each other, while the **G** components are markedly more correlated with each other in the European dataset than in the megacity datasets, with biological SOA being the exception.

## Appendix C: Concentration and uncertainty at selected m/zs for synthetic megacity ToF-ACSM OA data

Figure C1 shows example time series for selected m/zs from the synthetic megacity data.

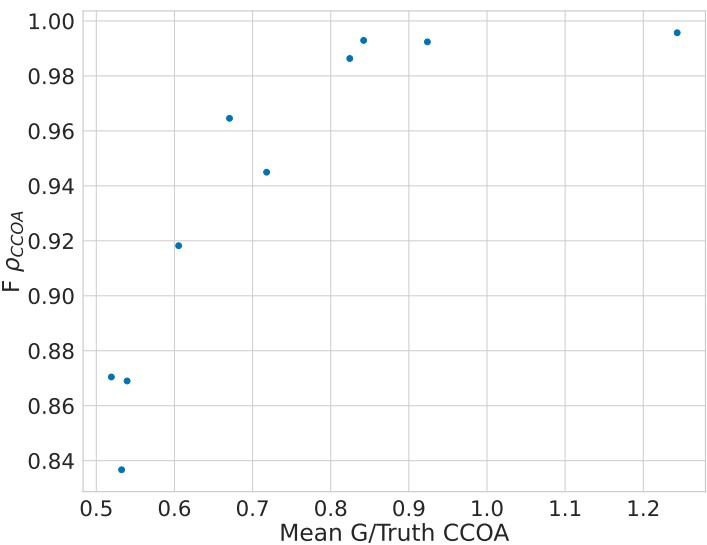

**Figure A2.** The bias of CCOA time series and composition in the 5 component megacity datasets.

**Table B1.** The Pearson correlation and standard deviation of the G components used to construct the 10 five component megacity datasets.

| Component | OOA | HOA | COA | BBOA | CCOA |
|---|---|---|---|---|---|
| OOA | 1 | -0.079 ± 0.067 | 0.039 ± 0.173 | 0.044 ± 0.158 | 0.033 ± 0.091 |
| HOA | -0.079 ± 0.067 | 1 | -0.076 ± 0.069 | -0.133 ± 0.104 | -0.171 ± 0.061 |
| COA | 0.039 ± 0.173 | -0.076 ± 0.069 | 1 | 0.330 ± 0.093 | 0.346 ± 0.047 |
| BBOA | 0.044 ± 0.158 | -0.133 ± 0.104 | 0.330 ± 0.093 | 1 | 0.503 ± 0.096 |
| CCOA | 0.033 ± 0.091 | -0.171 ± 0.061 | 0.346 ± 0.047 | 0.503 ± 0.096 | 1 |

**Table B2.** The Spearman correlation of the F components used to construct the five component megacity datasets. Note that F does not change between datasets, so standard deviation is 0.

| Component | OOA | HOA | COA | BBOA | CCOA |
|---|---|---|---|---|---|
| OOA | 1 | 0.761 | 0.659 | 0.804 | 0.816 |
| HOA | 0.761 | 1 | 0.839 | 0.864 | 0.845 |
| COA | 0.659 | 0.839 | 1 | 0.850 | 0.754 |
| BBOA | 0.804 | 0.864 | 0.850 | 1 | 0.871 |
| CCOA | 0.816 | 0.845 | 0.754 | 0.871 | 1 |

**Table B3.** The Pearson correlation of the G components used to construct the European dataset.

| Component | HOA | BBOA | SOA traffic | SOA bio |
|---|---|---|---|---|
| HOA | 1 | 0.773 | 0.718 | 0.005 |
| BBOA | 0.773 | 1 | 0.629 | 0.073 |
| SOA traffic | 0.718 | 0.629 | 1 | 0.018 |
| SOA bio | 0.005 | 0.073 | 0.018 | 1 |

**Table B4.** The Spearman correlation of the F components used to construct the European dataset

| Component | HOA | BBOA | SOA traffic | SOA bio |
|---|---|---|---|---|
| HOA | 1 | 0.852 | 0.844 | 0.801 |
| BBOA | 0.852 | 1 | 0.875 | 0.847 |
| SOA traffic | 0.844 | 0.875 | 1 | 0.845 |
| SOA bio | 0.801 | 0.847 | 0.845 | 1 |

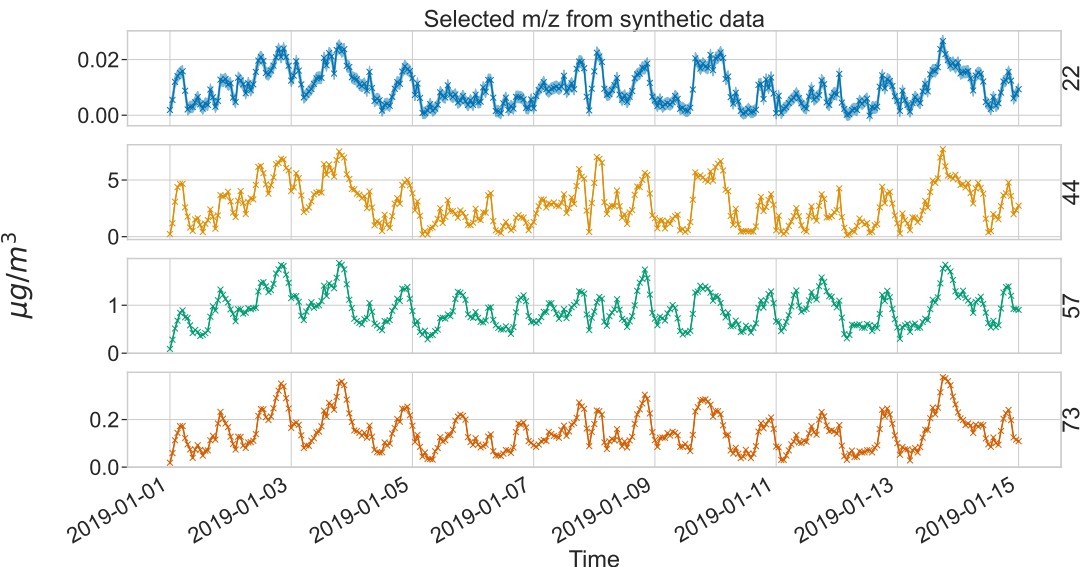

**Figure C1.** Concentration and uncertainty time series at selected m/zs for synthetic megacity ToF-ACSM OA data. The shaded area contains 95% of the probability mass of the Gaussian distribution of the error.

## Appendix D: Overspecified BAMF-C results

Figure D1 shows results from BAMF-C with too many components.

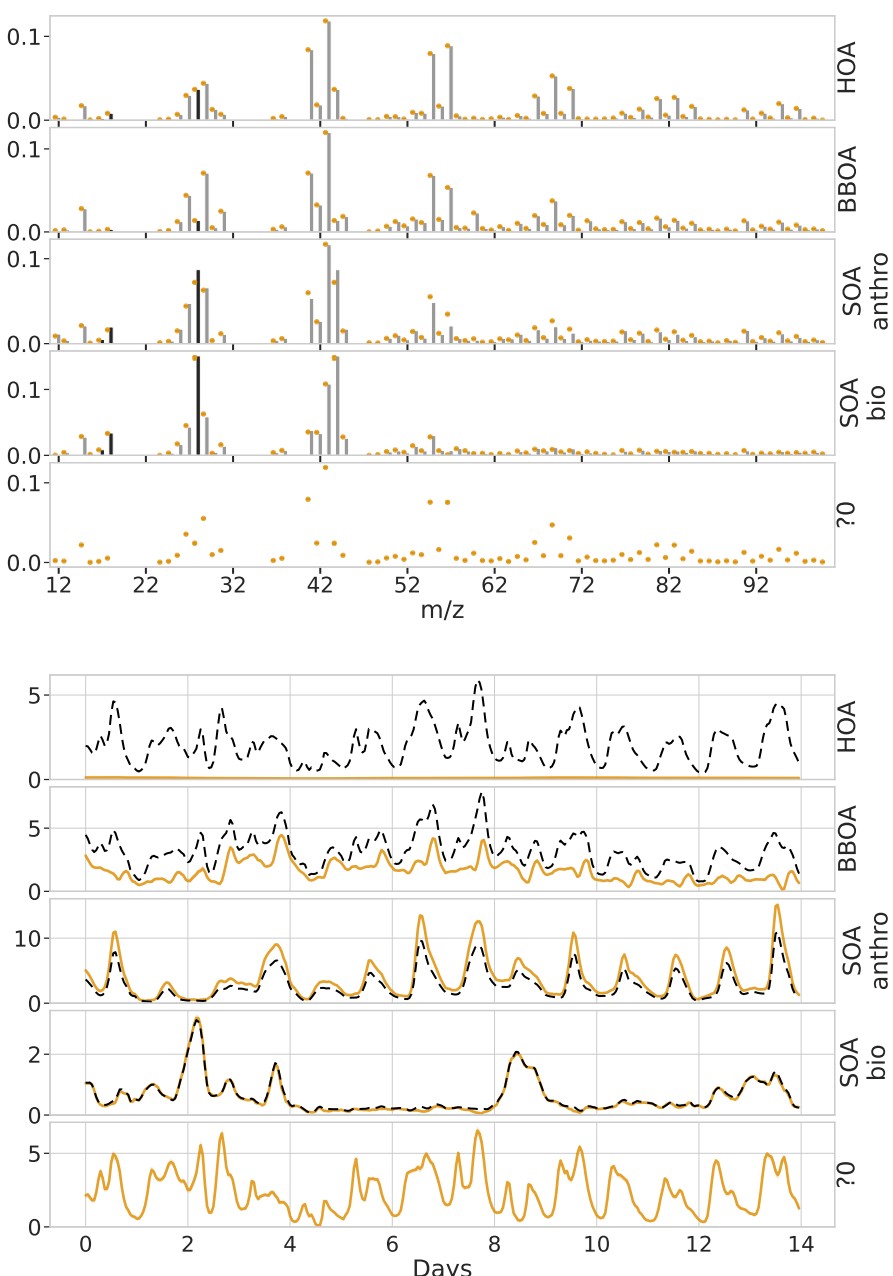

**Figure D1.** Results from overspecified BAMF-C model for the synthetic European city ToF-ACSM OA dataset with 5 modelled components instead of 4 and HOA & BBOA fully constrained.

## Appendix E: Workflow in this study

Figure E1 summarizes the steps used to run the BAMF model in this study and their order. In this study the profiles were from literature (see Appendix F). In general use the data generation step is not needed.

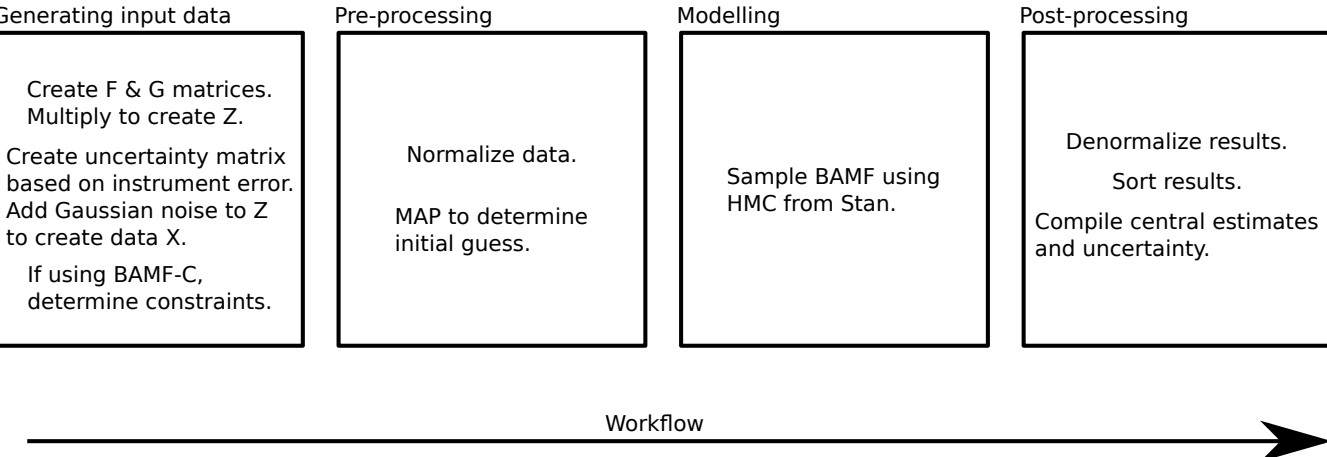

**Figure E1.** Workflow of running the BAMF model in this study. Pre- and post-processing steps are technically optional, but help in the convergence and interpretation of the results. With PMF the preprocessing and denormalization are skipped and the modelling box is just PMF, but we still sort the data similarly.

## Appendix F: Used profiles and constraints

Table F1 summarizes the used source profiles and constraints. Constraints were only applied in the European dataset.

**Table F1.** The source profiles used to construct the datasets. The profiles were restricted m/z 12 to 100, summed to unit mass intervals, and normalized to sum to 1. * indicates the values from this profile were also used as constraints.

| Component | Megacity dataset | European dataset |
|---|---|---|
| HOA* | Elser et al., (2016) | Elser et al., (2016) |
| BBOA* | Elser et al., (2016) | Elser et al., (2016) |
| CCOA | Elser et al., (2016) | Not applicable |
| COA | Elser et al., (2016) | Not applicable |
| OOA | Elser et al., (2016) | Not applicable |
| SOA bio | Not applicable | Daellenbach et al. (2017) |
| SOA anthro | Not applicable | Sage et al. (2008) |

## Appendix G: Comparison of constrained PMF and BAMF

Figure G1 compares PMF with constraints to BAMF without them. This represents the absolute best case scenario for PMF where you know the exact HOA and BBOA profiles beforehand. As mentioned in the main text, this does not fix the inability to resolve SOA bio.

## Appendix H: Error estimates and interquartile range

The error bars and the shaded areas in the time series are based on the interquartile ranges (IQR) in the empirical distribution given by the MCMC sampler. This gives us an idea of how accurately we can fix the modelled concentrations and compositions. In these results the error estimation is a bit optimistic, since it does not always cover the true solution. The underestimation is possibly due to the strictness of IQR and it not considering the model choice error. Figure H1 shows how the IQR compares to the median answer, with small concentrations having the most relative uncertainty. It also shows that BAMF-0 and PMF are often more uncertain than BAMF. In the case of PMF this is probably due to the values not being samples but solutions with different random seeds.

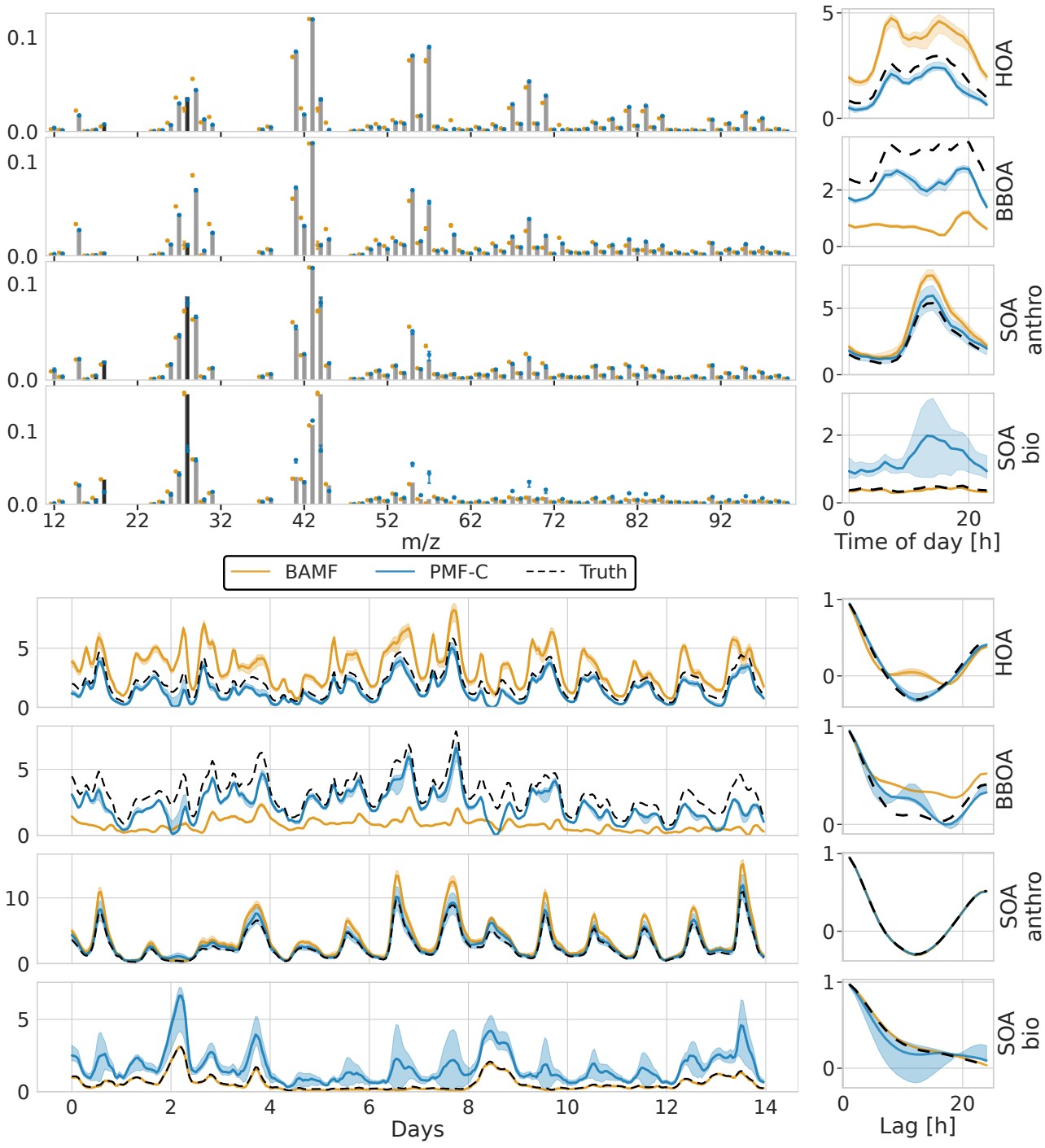

**Figure G1.** Unconstrained BAMF and constrained PMF on the European dataset with 4 components.

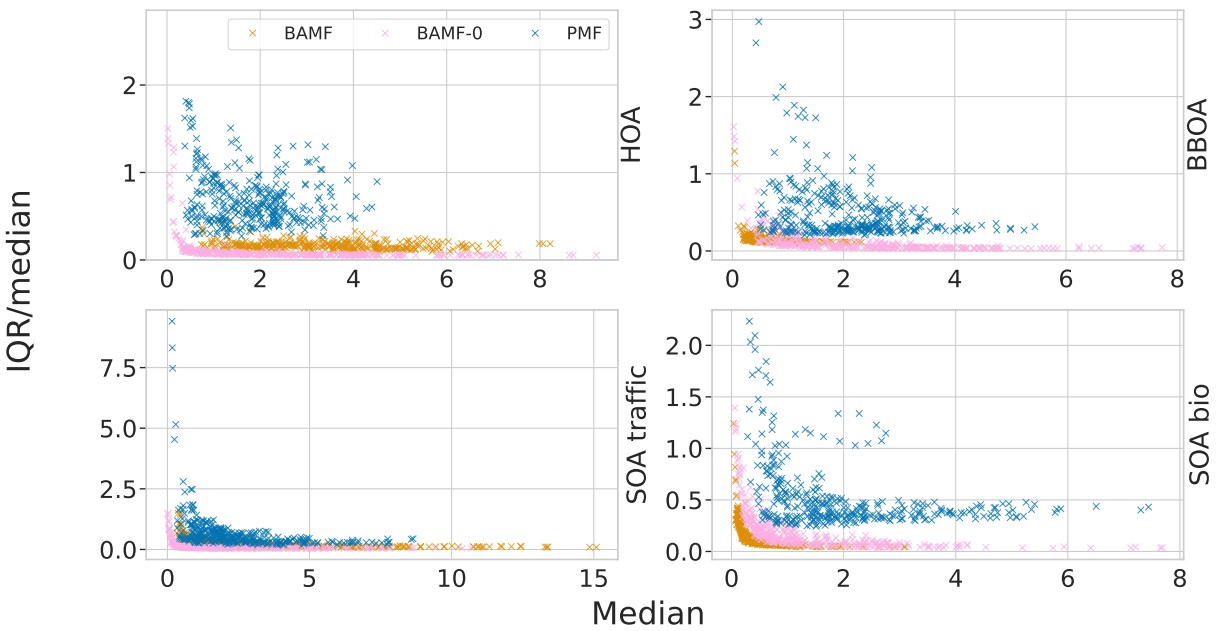

**Figure H1.** IQR compared to the median on the base case of the European dataset

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
