# Peer review of "A novel probabilistic source apportionment approach: Bayesian Auto-correlated Matrix Factorization"

_Atmospheric Measurement Techniques, 2023_

## Author Comment (AC1)

The referees' comments are in **_bolded cursive font,_** and ours are in normal font.

**Referee 1**

    **_General comments_**

    **_The manuscript by Anton Rusanen et al. constructed and evaluated a source apportionment (SA) methodology comprehensively and systematically. The Bayesian Matrix Factorisation (BAMF) model assumes the autocorrelation of PM sources, hence, it is compared to models in which autocorrelation is not assumed, such as Positive Matrix Factorisation (PMF), which is the most widely used. Also, it presents an a priori constraint effectivity assessment. Because of the worth of such a thorough exploration of a novel and advantageous SA mindset, I recommend publishing the manuscript subject to the revisions listed below._**

    **_My major concern is the lack of clarity in the manuscript due to the omission of details and discussion. Although the article is well structured, it misses information which would help the reader to understand the reasons for the decision-making. Moreover, the discussion of results is incomplete, and the conclusions are rather succinct and weak despite the strong results found. The manuscript needs to be thoroughly revised to show extensively the beneficial points of the BAMF and outline its limitations through a deeper explanation of the findings, model and factor-wise. The specific comments below list some points in which the explanations are weak or nonexistent._**

Thank you for the assessment of our work. Considering the points raised by the reviewer, we have added more details and deeper discussion based on your and the other referees' extensive comments on all aspects of the manuscript. The major changes are strengthening the evidence on auto-correlation, clarifying used model assumptions and constraints and collating much of the information in the appendices, please see the responses below for more details.

    **_MAJOR COMMENTS_**

    **_1. One of my major claims is the unreferenced statement that PM sources are autocorrelated, there is not any reference supporting it throughout the text. It is not unlikely, but this statement needs compelling backing arguments._**

We changed Figure 1 to include several measurements covering the major chemical PM2.5 constituents (BC, OA, NO3, SO4, Chl, NH4) from 19 different European measurement sites. This data shows that particulate matter constituents exhibit strong lag-1 autocorrelation, consistent with earlier such statements in literature (e.g., Hirtzel 1982).

We added discussion:

"This data shows that particulate matter constituents exhibit strong lag-1 auto-correlation, consistent with earlier such statements in literature (e.g., Hirtzel 1982)."

> *Line 44: This statement is too strong to be only justified by a time series of a certain site and period. You should back up this with references or more proof which highlights the need of considering auto-correlation in PM measurements. Also, further explanation on the cyclicity expected in 24h and its multiples and maybe intra-day correlation should be provided. Which sources are more (and less) expected to present such autocorrelations?*

*We changed Figure 1 to show multiple measurements showing a degree of autocorrelation. We added text with references about cyclicity:*

"Earlier studies have also found sources with longer cycles due to emissions, such as traffic, and meteorological conditions (Daellenbach et al., 2020; Chen et al., 2022). "

> *Sine the BAMF is a newly-developed SA method, the reader might need more information about it to fully understand its mechanism than that provided in Section 2.2. Could you please provide further information on how the algorithm works and which steps it follows? Maybe you can add a workflow diagram to clarify the process.*

We created a new subsection (2.3.1) with an overview of the algorithms and further references:

"Stan (Carpenter et al., 2017) uses a Hamiltonian Markov Chain Monte Carlo method to draw samples from the posterior distribution of our model (Equations 1-4) given the data. We go through the basic idea here, but direct readers to (Carpenter et al., 2017; Gelman et al., 2014) and references therein for a more detailed explanation.

The samples are drawn in proportion to the posterior probability of each sample. Obtaining samples from a multidimensional posterior distribution is a non-trivial task. For effective sampling, we use Hamiltonian Monte Carlo (HMC), a method where the gradient of the distribution and an ancillary variable called momentum are used to direct the chain to explore the typical set (Gelman et al., 2014). Specifically, we use the HMC based No-U-Turn Sampler (NUTS) sampler (Hoffman and Gelman 2014) from Stan.  Stan uses warmup iterations to estimate the parameters the NUTS sampler needs before drawing the posterior samples used in the computations (Carpenter et al., 2017).

The maximum-a-posteriori (MAP) estimate is used as a starting point for the sampling. It is found with an optimization method on the same probability distribution used by the sampling. We use the LBFGS (Liu and Nocedal, 1989) gradient-based optimization algorithm included in Stan for MAP estimates (Carpenter et al., 2017)."

 We also added a workflow diagram in Appendix E.

[Figure]

| Generating input data | Pre-processing | Modelling | Post-processing |
|---|---|---|---|
| Create F & G matrices. Multiply to create Z. Create uncertainty matrix based on instrument error. Add Gaussian noise to Z to create data X. If using BAMF-C, determine constraints. | Normalize data. MAP to determine initial guess. | Sample BAMF using HMC from Stan. | Denormalize results. Sort results. Compile central estimates and uncertainty. |

Workflow

*Line 60: "… measurement uncertainty determined as standard deviations of the error terms for each data point." Why did you not calculate the uncertainty matrix as widely calculated, in the way described in the protocol of Ulbrich et al. (2009)? A different uncertainty matrix calculation could significantly affect model performance.*

BAMF relies on a data matrix and an uncertainty matrix. Both together define for each m/z and point in time the distribution of measured values within its error. In this, the data matrix describes the average and the error matrix, the standard deviation of the distribution.

We determine the uncertainty matrix using the standard approach for ToF-ACSM, relying on Allan et al.(2003) and Ulbrich et al. (2009) (see section 3). We corrected the misleading sentence, which now reads:

"Together they define the probability distribution of the observed concentration of each m/z at any point in time with the data matrix being the average and the uncertainty matrix the standard deviation of the distribution, here the Gaussian distribution."

*The autocorrelation term implies that, from the Cauchy distribution, for high time lags (dt) the autocorrelation for the i, and i+1 time series is a uniform distribution. This might be true for certain sources, however, the traffic source, for instance, is expected to correlate better for dt = 24h rather than for dt=16h. Did you consider this drawback? How detrimental do you think the cyclicity disregard can be?*

In the present manuscript, we only use lag-1 autocorrelation, i.e., the autocorrelation of observations neighbouring in time (Line 67, original manuscript), while we acknowledge that the autocorrelation formulation deserves attention in future research. We note that each added lag autocorrelation term increases the required computation. Based on the new Figure 1, lag-1 ACF is a good assumption. In comparison, only some sources, as the reviewer points out, traffic, have a pronounced ACF also at other lags. We added a sentence to clarify that we only consider lag-1 auto-correlation:

"It is also possible to consider more than lag-1 auto-correlation."

Missing data can be handled in a lag-1 autocorrelation model by two approaches. Either these points are assumed to be not related at all or that larger the time difference the larger the difference in concentration. Here, we opted for the second.

> *In all discussions of plots it is lacking certain discussion on which factors are more accurate in each method and why a model might achieve a better description than the other. For example:*

> *Figure 4. BAMF does a better job overall, but maybe PMF does better for CCOA, can you hypothesise this? Relate to Table 1 results. Also, highlight that those factors with marked cyclical autocorrelation are those in which BAMF overperforms significantly BAMF-0 (e.g. COA).*

PMF has slightly lower G/Truth for CCOA, but otherwise the metrics in Table 1 indicate it is not better. Figure 4 shows that this is due to it over- and underestimating at different times, while BAMF & BAMF-0 have a constant slight overestimation, while correlating extremely well.

We added sentences describing this:

"All models slightly underestimate OOA, which results in overestimating of the other components (Table 1). For the final component, CCOA, PMF has less bias, but it correlates worse with the truth. BAMF-0 and PMF significantly underestimate OOA, which results in an overestimation of other components to get the reconstruction correct. Some factors have a more pronounced cyclical temporal behaviour as witnessed by high autocorrelation at higher temporal lag (e.g., Fig2c CCOA, BBOA, COA). Based on Table 1, some of the factors with pronounced cyclical temporal behaviour are better represented by BAMF than PMF (e.g., BBOA, COA), while this is not necessarily the case for others (e.g., CCOA)."

> *Figure 5: Why do you think the CCOA is the hardest factor to resolve even if the BBOA is well-described? It is agreed that it might be mixed with the BBOA, but what do you hypothesise makes BAMF-0 > BAMF?*

As BBOA is underestimated, CCOA is overestimated (appendix A) this would indicate these two components are hard to separate. BBOA & CCOA have almost identical F profiles (see appendix B) and have decreasing bias as F is better resolved (appendix A) as well as very similar time behaviour (peaks at the same time in diurnal and high correlation in G, see appendix B). This means the solution is very similar when you remove mass from one to and put it to the other. One can see that the range of G/Truth for BBOA and CCOA goes both sides of 1 for BAMF. BAMF-0 does markedly better in F rho for CCOA than BAMF, since the spread of BAMF is larger. We think this is due to mixing between CCOA and BBOA. Overall mixing is a major issue for all models, since the F profiles are very similar.

BAMF produces timeseries that are more correlated with the true factor time series than BAMF-0, throughout, including CCOA. However, the reviewer is right that the relative bias in the average CCOA concentration is larger for BAMF than BAMF-0 as well as the factor profile less well resolved. This suggests that the improvement in the temporal behaviour reduces the performance in recovering the factor profile which is in turn related to the mass concentration. Appendix A, Figure A2, shows that CCOA mass concentration bias increases quickly as F reconstruction degrades.

We added explaining sentence to the main text:

"Appendix A shows the inverse relationship between CCOA and BBOA mass concentration biases and how the profile reconstruction affects the CCOA mass concentration bias for the BAMF model."

We added appendices A & B to explore the possible reasons as well as elucidate the correlation of the components in both datasets.

> ***Figure 6: Discuss why the POA is better acknowledged by the PMF and SOA by the BAMF. Even if the final values shown in Table 2 support BAMF, don't you think the POA bias is significantly high? Also, discuss in terms of difference depending on the lags factor-wise.***

We agree with the reviewer and acknowledge in the main text that all models are challenged by this synthetic dataset.

We added text in the results section:

"All models show signs of mixing between the components, likely due to the correlation of the true G components (time behaviour is very similar) as well as similar chemical signatures F. PMF mixes the SOA components while BAMF mixes the POA components. However, it is worth noting that BAMF has a significant bias on several components in G as seen in Table 2, but otherwise reflects their time behaviour well."

And in conclusions:

"Both BAMF and PMF are challenged by strongly correlated components (European data)."

As the reviewer describes BAMF exhibits biases in certain components, however, the POA/SOA ratio (0.97) is well captured compared to PMF (0.78). Given the good performance in separating POA and SOA, BAMF likely mixes between BBOA and HOA explaining that HOA is overestimated and BBOA underestimated.

We added this to the main text:

"However, it is worth noting that BAMF has a bias on several components in G as seen in Table 2, but otherwise reflects their time behaviour well. Given the recovered POA/SOA ratio, BAMF likely mixes HOA and BBOA explaining that HOA is overestimated while BBOA is underestimated."

> ***Figure 9: Highlight the lag differences to truth of all models factor-wise.***

We added further content to the main text describing Figure 9d and compared it to Figure 6.

"Figure 9d shows that PMF-C captures HOA time behaviour very well, while BAMF still overestimates some auto-correlation, even though the bias in Figure 9a is significantly reduced. PMF-C also has some solutions that start to approach correct SOA bio, with solutions falling between unconstrained PMF and BAMF-C. For BAMF-C and BAMF SOA bio is virtually unchanged (Figures 9a, 9d, 6a and 6d)."

> ***Please justify the intermediate approach for constraining profiles used in section 4.4. Why would you constrain half spectra? Why did you select m/z60 as the threshold?***

Knowledge on the entire factor profile is not always available, e.g., not the same m/z range is measured (incomplete constraint), some ions are present/absent (partial constraint), etc. Thus, the goal of the intermediate approach is to test whether a priori knowledge on parts of the factor profile improves the

factorization performance. The split point in our experiment (m/z 60) was arbitrary as it was used mainly to show that this is possible with the model and even limited amount of a priori knowledge on the factor profile improves the factorization performance (See Figure 10).

We added this information to the main text:

"Incomplete constraint: Knowledge on the entire factor profile is not always available, e.g., not the same m/z range is measured. With the incomplete constraint approach, we test whether a priori information on parts of the factor profile improves the factorization performance and thus whether such information is useful. For the BAMF model, a priori information was only used in a limited arbitrarily chosen m/z range (12–60) instead of for all m/zs (m/z 12–100), same allowed deviation from the anchor for the constrained components.

Partial constraint: Sometimes, only very little chemical information is available for a specific factor, and it is defined by few key tracers. With the partial constraint, we test whether a priori information on just a few peaks/variables improves the factorization performance. For the BAMF model, a priori information was only used for 4 arbitrarily chosen m/z peaks out of 74 (m/zs 45, 57 and 60, F[ii,jj], compared to m/z 43, F[kk,ll]) for HOA & BBOA defined with the same allowed deviation from the anchor for constrained components. "

> ***Discuss how these two datasets are representative of real-world measurements and how reproducible are these tests in other datasets of different locations, the assortment of sources, temporal variabilities, meteorological influence… Is there any case in which you would rather apply PMF?***

In the present study, we assessed the performance of BAMF on two types of synthetic data that mimic the characteristics of real-world OA ToF-ACSM data as realistically as possible (e.g., error matrix calculation). We chose two types of synthetic ToF-ACSM OA datasets differing in their construction as well as setting (polluted megacity vs typical European city) reflecting the sources affecting such locations (HOA, COA, BBOA, CCOA, OOA and its separation). Generally, ACSM profiles are similar in most locations and the same constraints are used in many studies. Other measurements might be more varied. At this stage, wider testing including on other data types (e.g., trace elements from Xact) and real-world data remains. Nevertheless, we could show that if factors/sources are autocorrelated, then considering this feature improves the source identification, as done with BAMF.

We added text indicating this in conclusions.

"Other possible topics are using BAMF with other time series instruments and with real world data."

> ***Discuss how suitable would be BAMF for other instruments apart from ToF-ACSM (Q-ACSM, AMS, offline filters, PTR-MS, X-ACT…).***

While BAMF is tested presently on synthetic OA ToF-ACSM data, accounting for PM sources autocorrelation when performing source apportionment via matrix factorization is beneficial also for other data types such as Xact and offline filter data if the components are autocorrelated. Other ACSM and ASM data is likely to have similar behaviour. Further testing is especially needed for datasets with temporally sparse sources, i.e., pollution sources occurring during specific events. However, PMF is also

challenged by such data. The essential consideration is to balance a potentially improved factorization performance in comparison to the required additional computational time/power.

We added such content to the conclusions section:

"While we presently test BAMF on synthetic OA ToF-ACSM data, source apportionment analyses of other chemical PM data (e.g., trace elements from either Xact or offline filter analysis) could also profit from accounting for the auto-correlation of components, if the components are auto-correlated. Further testing is especially needed for datasets with temporally sparse sources, i.e., pollution sources occurring only during specific events, which are also challenging for PMF. "

> *You mentioned in Line 5 how BAMF provided error estimation. Please, discuss the uncertainties provided by the model for each factor.*

We added discussion and a Figure about the errors and their magnitude in appendix H:

"The error bars and the shaded areas in the time series are based on the interquartile ranges (IQR) in the empirical distribution given by the MCMC sampler. This gives us an idea of how accurately we can fix the modelled concentrations and compositions.  The error estimation is a bit optimistic, since it does not always cover the true solution. The underestimation is possibly due to the strictness of IQR and because it does not consider model choice error. Figure H1 shows how the IQR compares to the median answer, with small concentrations having the most relative uncertainty. It also shows that BAMF-0 and PMF are often more uncertain than BAMF. In the case of PMF this is probably due to the values not being samples but solutions with different random seeds."

> *Discuss if you believe the positive effect of considering factor autocorrelation (BAMF) is higher than the time-dependency of profiles effect (accounted by rolling PMF).*

Here, we use data equivalent to 2 weeks. This is roughly in line with the amount of data typically used in 1 data window for performing rolling PMF. Thus, BAMF could be used as rolling BAMF equivalent to rolling PMF. While this should be addressed in future research, it is beyond the scope of our manuscript.

We added a sentence to conclusions that using rolling window method would be an avenue for further research with the model:

"One such research topic would be introducing rolling window methods as has been done with PMF, to allow the source profiles to change over time and to act as a basis for real-time source apportionment."

> *Conclusions are short and weak despite the great number of tests performed and the valuable results obtained. Please elaborate on your findings:*
>
> *Improvements of BAMF for PMF on factorisation performance and limitations on reconstruction performance.*

We adapted the conclusion including more discussion on the factorization and reconstruction performance:

"All models performed well in reconstruction performance regardless of factorization performance, indicating that reconstructing the data is insufficient to judge how good the extracted factors are. Without strongly correlated components, BAMF resolves temporally autocorrelated components well (synthetic megacity dataset), while PMF performs considerably worse. Both BAMF and PMF are challenged by strongly correlated components (European data)."

> ***Results on the anchoring. Show which approach showed better results in BAMF-C.***

We added the following content to the conclusions:

"Even adding a priori information for a few peaks significantly reduced component bias, and partially specifying the profile (for 56% of the peaks) produced comparable results to fully constraining the profile with PMF. This opens up possibilities for using incomplete chemical composition information to improve factorizations."

> ***What are your intentions and further steps for BAMF? Is it prone to substitute PMF in the near term? Which aspects of the BAMF need future research?***

We commented on a key research topic that should be addressed:

"Overall, we believe BAMF-type models are promising tools for source apportionment and deserve further research, e.g., improving the separation of the chemical composition of components or the computational speed of BAMF. These models can also be used complementary to current source apportionment methods due to their different emphasis and advantages."

> ***MINOR COMMENTS***

> ***Line 2: "… are temporarily auto-correlated". I believe this sentence is very strong if unreferenced. Maybe it could be better to point out how this has never been accounted for when designing source apportionment methods.***

As mentioned above, we added more compelling evidence, in Figure 1, that PM and its components (measured data from 19 locations in Europe) are autocorrelated. We also added a reference to Hirtzel et al. (1982).Line 6: "…better than PMF…". "Better" here sounds arbitrary unless you present some proof showing how and why this model is better. Maybe you could present the Pearson correlation coefficient for Gs comparing them with PMF, or something similar referring to the factorisation performance.

We included more specific information in the abstract:

"We find that BAMF resolves sources with overall higher factorization performance (temporal behaviour and bias) than PMF on all datasets with auto-correlated components. Highly correlated components continue to be challenging and ancillary information is still required to reach good factorizations."

> ***Line 7: "highly cross-correlated components": since this has not been properly explained yet, I would advise giving further information about which are these components or substituting the expression with something like "not auto-correlated" or "susceptible to mixing". Also, give a reason for these factors to be challenging to resolve.***

We refer to components' time series or factor profiles being correlated. We changed the misleading formulation "cross-correlated" to "correlated" here and in all other places where cross-correlation term was used:

"We find that BAMF resolves sources with overall higher factorization performance (temporal behaviour and bias) than PMF on all datasets with auto-correlated components. Highly correlated components continue to be challenging and ancillary information is still required to reach good factorizations."

> ***Line 16: SOA is not only a result of gas-to-particle reactions, there are other formation mechanisms such as the coating of pre-existing particles. Hence, this sentence has to be rephrased or completed.***

We completed the sentence:

"… is distinguished from OA formed in the atmosphere from emitted vapours, by nucleation or condensation (secondary OA - SOA)."

> **Line 22: Please include the number of citations of the PMF paper, statistics on articles using this method, or reference to some paper which shows the predominance of this methodology.**

We reformulated the statement and added a reference that provides a number of PMF studies:

"A widely used method in atmospheric sciences is positive matrix factorization (PMF) (Paatero and Tapper, 1994), which has been used in over a thousand papers (Hopke, 2016)."

> ***Line 26: The paper from Canonaco et al. (2013) does not "optimise the source profiles", it demonstrates the high variability of profiles year-wise and suggests a method for accounting for certain capture of these evolving profiles (seasonal PMF). Please rephrase this sentence.***

The reviewer is right that Canonaco and colleagues published a paper in 2021 introducing a way to account for the temporal variation of profiles (Canonaco et al., 2021), while Canonaco et al. (2013) introduces Sofi (an IGOR-based interface for the efficient use of the generalized multilinear engine (ME-2) for the source apportionment: ME-2 application to aerosol mass spectrometer data) and the use of a priori information on the profiles.

We refer now to rolling PMF in the following statement:

"While developments related to source apportionment focused on different ways to pre- and post-process data (Zhang et al., 2019; rolling PMF: Canonaco et al., 2021), in atmospheric science, the underlying solver algorithm mainly remained the same, PMF."

> ***Line 27: "point solution with arbitrary rotations". PMF accounts for rotational ambiguity and statistical error assessment, hence, it does not provide a "point solution". Maybe I did not understand what "point solution" refers to since this concept is not explained here nor before. Please add some explanation on its meaning and why PMF is providing "arbitrary rotations" even if the nxm space is explored with rotational tools (e.g. a-value, random seed, DISP etc.)***

We removed this confusing sentence.

> ***Line 33: Add the citation of Heikkinen et al. (2020) which struggles with SA of the SMEAR II site low concentrations but managed to obtain main sources using machine learning techniques.***

We added the suggested reference:

"However, distinguishing factors with chemical or temporal similarities or accurately resolving low-concentration factors is often challenging (Ulbrich et al., 2009; Canonaco et al., 2013; Zhang et al., 2011; Heikkinen et al., 2021)."

> **Line 34: "…constrain POA sources chemical composition is usually…". Here, it should be mentioned that some sources were better captured using constraints applied in time series, as in Chazeau et al. (2022).**

Based on the reviewer's suggestion we changed the sentence:

"Several studies have shown that utilizing a priori information to constrain POA sources' chemical composition or time series is usually required to accurately estimate their contribution to OA (Canonaco et al., 2013; Crippa et al., 2014; Reyes-Villegas et al., 2016; Schlag et al., 2017; Zhang et al., 2018; Huang et al., 2019; Zhu et al., 2018; Chazeau et al., 2022)."

> **Line 41-42: "…the commonly used optimization goals do not include any temporal terms of the resolved components (Wang and Zhang, 2012; Paatero and Tapper, 1994), and thus any time information is ignored.". The reader might find it hard to understand what this refers to. Do you mean that the optimisation parameter Q in PMF is not related to X temporal features? Please rephrase.**

We rephrased the sentence for a better readability:

"The commonly used optimization goal Q in PMF only accounts for reconstruction of the data (Wang and Zhang, 2012; Paatero and Tapper, 1994). It lacks time information, which is a drawback because some atmospheric measurements exhibit strong temporal auto-correlation (see e.g., Figure 1)."

> **Line 54: "Simply put… time-dependent concentration". Here you could indicate that this implies the staticity of the profiles throughout the data, and how PMF has overcome this issue with the rolling PMF, achieving time-dependent profiles.**

Rolling PMF refers to a pre-processing strategy feeding only chunks of data (e.g., 7 or 14 days) to the PMF solver. This allows for a temporal variation of the profiles even if they remain static within each PMF run. BAMF is entirely compatible such an approach and thus can be used for rolling BAMF which is though out of the scope of the current study. We added this content to the introduction section of the manuscript:

"While developments related to source apportionment focused on different ways to pre- and post-process data (Zhang et al., 2019; rolling PMF: Canonaco et al., 2021), in atmospheric science, the underlying solver algorithm mainly remained the same, PMF. Rolling PMF refers to a pre-processing strategy feeding only subsets of data (e.g., 7 or 14days) to the PMF solver. This allows for a temporal variation of the chemical composition of sources (particularly relevant for SOA), even if their profiles remain static within each PMF run (Canonaco et al., 2021)."

> **Lines 66-67: The reader might need more information on why these distributions are selected and what their parameters are needed. Firstly, the parameters of the distributions could be shown explicitly for greater clarity e.g. Cauchy (x0 = Gik, γ = αa[k]Δti +αb[k]). Also, the fact**

> *that Fi is a matrix of i dimensions should be mentioned. Does the F not represent now the*
> *factors' chemical composition? More information than that given in lines 71-75 is needed.*

Added indication of which parameter is location and scale for each distribution.

The why is described on lines 74-77 of the original manuscript.

The choice of distribution is arbitrary since it is used as an approximation of the auto-correlation process. There can be many viable approximations; we tested one.

The reason for Gaussian is the error description mentioned earlier in the manuscript (original manuscript lines 59-60):

"The error matrix contains the measurement uncertainty determined as standard deviations of the error terms for each data point."

This is the Gaussian distribution centred at the data point with a given standard deviation.

We clarified this point in another comment above relating to the error matrix.

The dot is crucial in the definition of $F_{I\_dot}$, which makes it a vector since it is a row of F. Indeed, it is the chemical composition of a factor since that is what F was defined as on line 52 of the original manuscript.

> *Line 77: "The term … determines…". Please explain explicitly the mathematical and*
> *environmental meaning of this term. Is it meaning that the lag of time implies less probably*
> *$G_{i+1,k}$ to be equal to $G_{i,k}$? Also, add information on the configuration and meaning of the*
> *vectors alpha.*

We added a sentence:

"In physical terms this means that at short timesteps we expect the values of G to stay close to the previous value and at large timesteps larger deviations have higher probability."

There is no deeper meaning to the vector's alpha and beta, it is a simple linear model. Amended the text to state this explicitly:

"The $\alpha_k \Delta t_i + \beta_k$ determines the scale of the Cauchy distribution. The $\alpha$-terms allow the model to deal with time steps of different lengths and missing data, since it forms a simple linear model for the scale."

> *Line 90: Equation (5). I don't think this can be understood with the information provided. Do*
> *you mean that you constrain each j and l $\forall$ j, l? Also, if I understand the intention of this*
> *equation, the ratio between two ions' intensities should be written as "F[i,j]/F[i,l]". Otherwise,*
> *you are expressing that you fix a ratio between m/zs for whichever times i, k, which would*
> *make no sense. However, it does make sense though constrain two ion intensities for all time*
> *units i. Moreover, in this fashion, you avoid using the k index for time, since it is inherently*
> *related to the number of factors for us the PMF users.*

The indices refer to literal elements of the matrix F. Matrix F has nothing to do with time. We changed the indices to ii, jj, ll, kk to avoid confusion with previous defined I,j and other indices.

Changed the description to:

"where ii, jj and kk, ll are the indices of matrix F indicating the peak pair to constrain"

> *Lines 90-99: You should specify which ions and from which reference profiles you are using here in the methods section. Please, refer to a table for indicating those.*

The description of the constraints is provided in detail on lines 306-311 of the original manuscript.

We have now added a table to Appendix F to clarify the constraint profiles and text indicating this to the methods:

"Appendix F lists the profiles used for constraints in this work."

> *Line 99: The SoFi PMF has a criteria selection to filter out bad values, hence environmental criteria to accept/discard solutions affects the final a-value, whose presented value will be the mean value of all the accepted runs. Hence, the appreciation you make of "all solutions within the boundary to be of equal quality" is not fair since only those solutions which make environmental sense are kept. Please, disregard this comment if I did not understand what you were intending.*

We reformulated the sentence that the reviewer highlights to improve clarity:

"SoFi/PMF employs a hard boundary (defined as a relative deviation from the a-value), without additional penalty on the object function Q for deviation from the anchor. In the present study, we evaluate the performance of the PMF and BAMF algorithm itself without discarding sub-optimal solutions (PMF) or samples (BAMF) during post-processing. "

> *Lines 109-112: The reader might not understand what you refer to with "MAP point solution", please describe a bit more this part.*

We added a brief description of what MAP is to the manuscript:

"The maximum-a-posteriori (MAP) estimate is used as a starting point for the sampling. It is found with an optimization method on the same probability distribution used by the sampling. We use the LBFGS (Liu and Nocedal, 1989) a gradient-based algorithm included in Stan for MAP estimates (Carpenter et al., 2017)."

 Using such an optimization approach is similar to how PMF acquires a solution but differs in technical details.

> *Lines 116-119: Why are you normalising to afterwards dis-normalise again? Can you please explain the motivation for this step?*

The idea here is to make the modelling easier since we don't need hyper-parameters to account for varying scales of the distributions. The normalization step is optional, as mentioned, and denormalization is done after the model runs simply to return to familiar units.

We added appendix E to clarify the workflow and a sentence to the description of normalization:

"This normalization is done so that we can use a consistent scale for priors and posteriors, making the modelling easier, and the denormalization is performed to return the results to familiar units. The normalisation is optional, a user can also choose to use non-scaled values. "

> *Line 132: you have to explain how you apply the Hungarian algorithm, Manhattan distances concepts here so that the reader understands the way it works specifically for sorting factors (at least).*

We agree with the reviewer, and we changed the text accordingly. We clarified that it is a cost minimization algorithm and distance is the cost. We clarified the Manhattan distance to be a sum of absolute differences:

"To select the ordering of the components, we take a small number of representative samples, usually the last five, and compute the optimal permutation using the Hungarian algorithm (Kuhn, 1955), which is a cost minimization algorithm which minimizes the cost of assigning values. In this case we are minimizing the Manhattan distances, which is the sum of absolute differences between the Z contributions in the samples. We then select the most common permutation as the ordering of the factors for all samples."

In depth explanation of cost minimization algorithms that can readily be found from scientific programming libraries is out of scope for the paper.

> *Lines 145-149: Why use this instead of scaled residuals? I know the meaning is analogous, but it surprised me that you computed those metrics instead of the commonly-used scaled residuals.*

Scaled residuals are used in Figure 3a. Median and maximum of scaled residuals are also shown as reconstruction performance in Tables 1 & 2. The metric in question is used to see if the true solution is within the modelled error estimate.

> *Lines 208-211: Please explain a bit more about the construction of G, why do you model it through random walks? How did you tweak the diel patterns through this kind of modelling? Moreover, I don't understand the sentence "the added diurnal peaks can be seen as the periodicity of the tail. Also, why BBOA is using a Gaussian and the other factors using Cauchy?*

We added clarifying sentences:

"Mix of different random walk distributions was chosen to test if the model approximation of Cauchy auto-correlation works with them. The random walk aims to introduce variability such as one would get from varying transport and mixing. The diurnal cycle was simply summed to the random walk to produce the time series."

We changed the confusing sentence to:

"The added diurnal peaks can be seen as the periodic peaks in the correlograms in Figure 2c and f."

BBOA is Gaussian because we wanted a mix of different behaviours. As mentioned before, there is more than one viable approximation for the time series.

> ***Table 1: Please justify why did you use r for G and ρ for profiles.***

We added a sentence:

"Nonlinear correlation coefficient is used for factor profiles, since they have an additional constraint of summing to unity and thus linear correlation is penalized for small errors disproportionately."

> ***Line 284: I would move these two graphs (Figs. 7 and 8) to supplementary information or annexe. These are not your final solutions but tests, which can conflict with the clarity of the solution presentation.***

Even though we acknowledge the referee's comment, we believe that determining the optimal number of factors is crucial and thus we consider the behaviour of BAMF and PMF with a sub-optimal number of factors important enough to be shown in the main text.

The splitting and combining behaviour of all the models when over/underspecified is something the user should be aware of, an area where BAMF can be argued to be an improvement and we consider it part of the main results. Your next comment does illustrate that we might not have emphasized this enough and the change is related.

> ***Line 289: Please describe the composition and time features of this "unnecessary factor". Discuss if it is more advisable in this respect to extract a "noise/unidentified" profile as the BAMF does or to split two factors as the BAMF-0 and the PMF do.***

To address these comments, we added the following sentences:

"For this component the time series is almost constant, and the composition is flat with large uncertainties"

"In general use, one would prefer the model to indicate the limits of the factorization as BAMF does, instead of producing duplicate components."

> ***Lines 317-318: "Partially constrained… ". Discuss which recommendation would you make in a ranking fashion: first the constrained PMF, but afterwards, the constrained PMF or the partially-constrained BAMF?***

We added a sentence ranking these:

"For this dataset and these constraints, BAMF-C fully constrained has the best factorization performance, fully constrained PMF and partially constrained BAMF-C are performing slightly worse but about equally good and unconstrained models fail to resolve one or more components correctly regardless of model."

> ***Figure 9 or 10: I would like to see a comparison of unconstrained BAMF vs. constrained PMF to assess the power of BAMF unconstrained vs. the best version of PMF, which is constraining it.***

We believe that a comparison between unconstrained BAMF and constrained PMF is not completely fair as it compares the best case of one model to the worst case of the other. Even though the goal would be to showcase the potential of BAMF, this comparison can still be misleading to the reader, as adding a priori information gives a huge advantage to any model. Nevertheless, we added such a figure to appendix G in addition to the information already presented in Table 2.

We added a sentence about this direct comparison:

"Such prior information is powerful, for example constrained PMF does better than unconstrained BAMF for all components except SOA bio."

> *Line 324: Again, these two first sentences are very strong and unsupported. Please add citations or reformulate the argument.*

We removed the sentences since they are not required to state the conclusions.

> *Technical corrections*

> *Line 41: "Particularly relevant for this study is that…". Please rephrase to something similar: "XX is particularly relevant for this study…".*

We rephrased the sentence to:

"The commonly used optimization goal Q in PMF only accounts for reconstruction of the data (Wang and Zhang, 2012; Paatero and Tapper, 1994). It lacks time information, which is a drawback because some atmospheric measurements exhibit strong temporal auto-correlation (see e.g., Figure 1)."

> *Line 49: Write the definition of j in the same way as the i definition, otherwise seems as if they were not analogous. It should be written as j $\in$ [m]={1, …, m}.*

We wrote out the definition.

> *Line 52-53: Why do you write Fi with a point? What does it mean? Is it completely necessary for the definition?*

Yes, as detailed above in the minor comments, it is crucial for the model definition and understanding for F_i_dot to be a vector.

> *Line 59: I believe the notation in PMF for the matrix error is called "uncertainty" rather than "error". Error would refer to the error matrix. Please, modify.*

We renamed the error estimate to uncertainty estimate.

> *Line 61: Please, define what you imply with the term "latent variables".*

It is defined in the next two sentences, all but X and sigma which are input or observed.

> *Line 83: "… without the lag-1 auto-correlation…". Could you rephrase? Do you mean that the Gi+1,k is not computed using the Cauchy distribution or that something in expression (4) is different for the BAMF-0?*

We added a clarifying sentence:

"In other words, the model consists entirely of Equations 1-3"

> *Line 88: Indicate the nature of this penalty and how is it applied.*

We changed the sentence to:

"In physical terms this means that at short timesteps we expect the values of G to stay close to the previous value and at large timesteps larger deviations have higher probability."

> **Line 98: This is not a scenario but a practice or a methodology. Please rewrite.**

We changed the last sentences to:

"Firstly, the intensity ratio of the peaks is constrained in BAMF-C, while the a-value constraint approach in SoFi/PMF uses the peak intensity. Secondly, BAMF-C has a soft-boundary, with increasing penalty term as distance from the anchor grows. SoFi/PMF employs a hard boundary (defined as a relative deviation from the a-value), without additional penalty on the object function Q for deviation from the anchor.

> **Line 101: Define the acronym STAN. Also, if not stated above, describe slightly the workflow of the algorithm/process.**

STAN is not an acronym as far as its documentation says, it is just the name of the program.

We changed the case to the one used by current documentation: Stan.

We included a brief description of the Hamiltonian Markov chain Monte Carlo process, as well as a workflow diagram (appendix E).

> **Beware that "reconstruction performance" (Line 140) is not in italics but these tests are in italics in line 56. Please homogenise.**

They are italics in the place where they are introduced to emphasize it is a specific term.

> **Line 169: Please describe what you mean by "sub-optimal".**

Not a minimum in terms of probability. Changed the sentence to:

"…, including plausible answers that are not minima."

> **In lines 186 and 207 you should be explicit about which anchor did you use for every factor.**

We have collated this information to Appendix F and added text directing the reader there:

"See appendix F for the profiles used to make this comparison."

> **Lines 205-206: "the modelled sources are traffic exhaust: HOA, cooking: COA, biomass burning: BBOA, coal combustion: CCOA, and secondary OA: OOA". The punctuation here is not used properly. Please, rephrase this sentence.**

We added semicolons:

"In our case, the modelled sources are traffic exhaust, HOA; cooking, COA; biomass burning, BBOA; coal combustion, CCOA; and secondary OA, OOA."

*Line 235: I believe this section should be called "Results and Discussion". The labelling "Experiments" could be implying a methodological explanation of them, which should not be the case.*

We renamed section as suggested.

*All graphs including profiles (Figs. 2, 4, 6, 7, 8, 9, 10) present an arbitrary x-axis ending in two, which does not look standardised. Please, include 12, but afterwards, use tenths' ticks.*

We agree with the reviewer that it is important to adapt the X axis to the data at hand. Thus, we start the axis at 12 and use regular ticks for clarity and in consequence refrain from switching to traditional ticks.

*Tables 1, 2: Captions are to be placed above the table not below.*

Based on the reviewer's input, we placed the caption above the tables.

*Line 310: Explicit which anchors did you use in each factor to constrain the ion ratios.*

We added notation which is the nominator and denominator and repeated that it is about HOA & BBOA.

"For the BAMF model, a priori information was only used for 4 arbitrarily chosen m/z peaks out of 74 (m/zs 45, 57 and 60, F[ii,jj], compared to m/z 43, F[kk,ll]) for HOA & BBOA defined with the same allowed deviation from the anchor for constrained components."

*References*

*Chazeau, B., El Haddad, I., Canonaco, F., Temime-Roussel, B., d'Anna, B., Gille, G., ... & Marchand, N. (2022). Organic aerosol source apportionment by using rolling positive matrix factorization: Application to a Mediterranean coastal city. Atmospheric environment: X, 14, 100176.*

*Heikkinen, L., Äijälä, M., Daellenbach, K. R., Chen, G., Garmash, O., Aliaga, D., ... & Ehn, M. (2021). Eight years of sub-micrometre organic aerosol composition data from the boreal forest characterized using a machine-learning approach. Atmospheric Chemistry and Physics, 21(13), 10081-10109.*

*Ulbrich, I. M., Canagaratna, M. R., Zhang, Q., Worsnop, D. R., & Jimenez, J. L. (2008). Interpretation of organic components from positive matrix factorization of aerosol mass spectrometric data. Atmospheric Chemistry and Physics Discussions, 8(2), 6729-6791.*

Citations

Hirtzel, C. S., et al. "Estimating the Maximum Value of Autocorrelated Air Quality Measurements." *Atmospheric Environment (1967)*, vol. 16, no. 11, 1982, pp. 2603–08.

Hopke, P. K.: Review of receptor modeling methods for source apportionment, Journal of the Air & Waste Management Association, 237–259, https://doi.org/10.1080/10962247.2016.1140693, 2016.

---

## Author Comment (AC2)

**Referee 2**

*The manuscript addresses source apportionment using a Bayesian statistical approach. This is a departure from standard source apportionment techniques and the approach has some conceptual merit in that it reduces reliance on measurement uncertainty matrix inputs and therefore facilitates the inclusion of additional parameters. The generation of a test dataset and the analysis using established positive matrix factorization and the novel Bayesian approach is helpful in move the science forward in this area.*

Thank you for the review. Please see our responses to specific comments below and in the answers to referee 1.

*For the wider application of this approach, it would be useful to understand what the computational speed is when compared to PMF.*

We added a consideration regarding the computational speed to the manuscript:

"Another area of development is computational speed, for the dataset sizes discussed here running BAMF takes few hours on a modern computer (Intel Xeon Silver 4110), but the time increases as the data size increases."

*For PMF, alternative factor solutions below and above that chosen should be reported and discussed at least in the SI*

The alternative factor solutions are already reported in Figures 7 & 8 as well as Table 2 and discussed in section 4.3 "Resolving an unknown number of sources", which was expanded due to a similar comment from referee 1.

*Fig 3 and associated analysis and discussion - it would be useful to have a statistical analysis of these comparisons (t-test, Kruskal-Wallace) to show whether the difference were statistically different*

Even though we acknowledge the reviewer's concern about the statistical significance of the differences, the main point here is that the distributions are very similar and all within limits that we consider acceptable.

*Fig 4 – there is a large over-estimation of HOA compared to the other approaches. It is not obvious where this mass is allocated in comparison and is worthy of some discussion.*

We added a sentence stating that it is probably taken from OOA (notice the order of magnitude difference in scales):

"All models slightly underestimate OOA, which results in overestimating the other components (Table 1)."

*Fig 5 – please keep the fig sub title (a,b,c) in the same location. The reason for the large variability in CCOA for BAMF in b needs to be discussed in more detail.*

We adapted the figure labelling as suggested by the reviewer.

We added a discussion about the mixing between BBOA and CCOA. The correlation of F is easily influenced by the removal and addition of mass between the two components. We added Appendix A and the sentence:

"Figure 5a, shows that BAMF can over- and underestimate both BBOA and CCOA depending on the dataset."

**First sentence of conclusion (326) is not true.**

We removed the first sentence of the conclusion.

**other comments:**

**Line 20 – formalism – what is this? Could an alternative word be used?**

We reformulated the sentence, which now reads:

"The idea is to use the variation in the chemical composition of a set of measurements, such as outputs from mass spectrometers, to decompose the measurements into "source terms" using non-negative matrix factorization."

**Line 66 - Dirilecht is spelt incorrectly I believe**

We corrected the spelling.

**Line 166 – Sofi only finds local optima for unconstrained PMF, where an a-value is used this is not the case**

To our best knowledge, using a-values does not change the nature of the optimization problem. It changes the pool of possible solutions.

**Citation:** **https://doi.org/10.5194/amt-2023-70-RC2**